# Investigating the Correlation of Tectonic and Morphometric Characteristics with the Hydrological Response in a Greek River Catchment Using Earth Observation and Geospatial Analysis Techniques

**Emmanouil Psomiadis [1,*]**, **Nikos Charizopoulos [1]**, **Konstantinos X. Soulis [1]** and **Nikolaos Efthimiou [2]**

1   Department of Natural Resources Management and Agricultural Engineering,
    Agricultural University of Athens, 75 Iera Odos st., 11855 Athens, Greece; nchariz@aua.gr (N.C.);
    soco@aua.gr (K.X.S.)
2   Faculty of Environmental Sciences, Czech University of Life Sciences Prague, Kamýcká 129,
    165 00 Praha, Czech Republic; efthimiou@fzp.czu.cz
*   Correspondence: mpsomiadis@aua.gr; Tel.: +30-210-529-4156

**Abstract:** Morphometric analysis can be used to investigate catchment dynamics and tectonic processes responsible for the development of drainage catchments and to support flood risk assessment. In this study, a comparative GIS-based morphometric analysis between the main southern and northern sub-catchments of the Sperchios River basin, Central Greece, was performed, using geospatial and remote sensing data. The goal was to investigate their correlation with the peculiar geotectonic activity and the frequent flash-flood events that occur in the river floodplain. All sub-catchments characteristics are linked with the geological formation types of the area, in combination with ongoing tectonic activity. The results indicate that drainage network development is significantly controlled by the region's overall tectonic activity. The morphometric characteristics—i.e., bifurcation ratio, drainage density, circularity ratio, elongation ratio and water concentration–time values, reflect the flood-prone character of the southern part of Sperchios River catchment in comparison to the northern part, especially during intense rainfall events. The study can provide valuable insight into identifying how morphometric characteristics are associated with increased flood hazard.

**Keywords:** morphometric analysis; tectonic activity; floods; GIS; remote sensing

## 1. Introduction

Morphometric analysis is the first step towards understanding catchment dynamics, providing a quantitative description of its drainage system, topographic features and other intrinsic attributes, such as shape, dimensions, etc. [1–3]. The development of a catchment's drainage network and the evolution of its morphological features are mainly controlled by lithology and structure [2,4]. Lithology affects the physical (e.g., permeability, susceptibility to erosion, hardness, etc.) and chemical (e.g., diagenesis, dissolution, etc.) properties of rocks, while structure impacts their geometry, composition and thickness, and the shift in their tectonic regime, which can generate faults and folds [5,6].

During the 1930s and 1940s, Horton's research [7–9], followed in the forthcoming years by the work of Strahler [1,10,11] led to the quantification of catchment geomorphology—i.e., the mathematical expression of its morphological characteristics. The quantitative description of geometric features at watershed scale can be used for illustrating its structural controls, geological and geomorphic history,

and drainage network processes [12–14]. Furthermore, the catchment's morphometric parameters play a crucial role in the ongoing hydrological processes, as they define, to a large extent, their hydrologic response [15–17]. The dynamic nature of runoff is controlled by the catchment's geomorphologic structure (shape, steep slopes, etc.) and is responsive to its morphometric characteristics. The knowledge of such behavior, specifically during intense rainfall events, is critical for flood hazard assessment, especially in ungauged catchments [17–20].

Additionally, recent findings have triggered new insights regarding the effect of geomorphological or tectonic factors on drainage systems and landscape development [12,21]. Neotectonics is a critical discipline in the field of landform evolution and catchment morphometry at tectonically active regions [22–24]. Earlier morphometric studies were based on traditional methods, such as field observations and topographic map "interpretations" [1,7,25]. Numerous researchers attempted to relate the tectonic activity with an area's morphometric characteristics and hydrological processes, utilizing quantitative analysis and morphometric indices, such as those of Mahala et al. and Babu et al. [26,27], which indicate the significance of morphometric and drainage catchment parameters analyses in order to understand the hydrological and morphological characteristics of an area, and those of Dehbozorgi et al. [21] Gao et al. [23], and Gaidzik and Ramírez-Herrera [24], which reveal the importance of tectonic active and uplift to the geomorphological evolution of a catchment [21,23,28]. A significant number of these studies took place during Greek hydrological watersheds, with similar tectonic and morphometric characteristics revealing the necessity of such investigation, such as those of Charizopoulos et al. [29], Argyriou et al. [12], Ntokos [30] and Kouli et al. [31], which demonstrate the impact of tectonic control in geomorphological and hydrological processes in Western Greece and on the islands of Samos and Crete. Likewise, Maroukian and Lagios [32], Paraschou and Vouvalidis, Eliet and Cawthorpe [33] and Pechlivanidou et al. [34] display the specific tectonic, morphometric and hydrological characteristics of the Sperchios river catchment.

Currently, however, the rise of Geographical Information Systems (GIS) and Remote Sensing (RS) technologies revealed new perspectives and led to more timely, cost-effective and accurate analyses [29,30,35–40]. GIS techniques are suitable for analyzing various morphometric parameters (terrain, drainage network, etc.), providing high computational capabilities for managing spatial information [41–43]. Moreover, recent RS data and Shuttle Radar Topography Mission (SRTM)-derived Digital Elevation Models (DEM) are used for the "extraction" of information regarding catchment's lineaments and geomorphological processes towards a detailed identification of tectonic and geomorphological features at catchment or at fault zone level [39,44,45]. DEM provides valuable insight into the fields of morphometric, tectonic, and hydrologic research [46,47]. SRTM-DEM is a unique source of medium to high resolution (30 m) data, mainly supporting morphometric and hydrological studies. The latter is based on the principle of interferometric SAR that utilizes different properties, deriving from two radar imageries acquired with a minor base-to-height ratio, in order to delineate topography [48].

Remote sensing images have a critical role in lineament (e.g., faults, joints, and dykes) mapping, due to the synoptic overview of outsized regions and their ability to deliver more accurate data of the Earth's surface, than those of coarse scale topographical maps [49,50]. For almost five decades, the NASA/USGS's Landsat program made available the lengthiest continuous satellite database of the Earth's surface [51–53]. Landsat data and multispectral remote sensing techniques have been extensively used for various geological and geospatial applications [54–56]. Landsat 7 (L7) ETM+ is appropriate for lineament extraction, since it delivers multispectral, high spatial resolution data that offer better insight into their detection [57]. In the past, lineament extraction research has used Landsat data and different ancillary techniques, such as Principal Component Analysis (PCA), gradient filtering and False Color Composites (FCC) [57–60]. Lineament recognition using RS data can be primarily carried out by applying two methods, comprising (a) visual interpretation and semi-automatic outlining, that depend on a general knowledge of the study area's geological background [61] and (b) automatic lineament mining that uses various computation parameters [57,62]. Automated procedures usually

provide additional objective lineament features which—in many cases—are difficult to distinguish from non-geological lineaments, such as roads, land use linear borders or other linear characteristics.

The overall objective of this study is to (a) identify the connection of the Sperchios watershed and sub-catchments' tectonic conditions of a catchment with their morphometry, by comparing the orientations of the tectonic faults and drainage network branches (use of rose diagrams), utilizing a combined quantitative and qualitative approach, and (b) assess the correlation of high tectonic control catchments (such as Sperchios) with faster runoff response time and more frequent flash-flood event occurrence.

## 2. Study Area

The study area lies at the Southern part of Sperchios River catchment, Central Greece, located between 38°44′-39°05′ North, and 21°50′-22°45′ East (Figure 1a). Sperchios River emanates from Mount Timfristos, having a West–East direction, forming a relatively large catchment, occupying an area of approximately 1823 km$^2$. The river runs through the valley developed between Mount Oiti and the Western extensions of Mount Othrys and Mount Kallidromo, outflowing to the Maliakos Gulf (Figure 1a,b). There is a linear correlation between rainfall and the altitude zones in the area, resulting in a precipitation variation from 400 mm in the lowlands to 1600 mm in the mountainous regions [42,63]. The mean annual runoff is 62 m$^3$ s$^{-1}$, ranging from 22 m$^3$ s$^{-1}$ in August to 110 m$^3$ s$^{-1}$ in January [64,65].

The Sperchios catchment is a graben-like asymmetrical depression as a part of a tectonic trough, controlled by major NW–SE and E–W trending faults parallel to the Atalanti normal fault zone [66,67]. The faulting took place at the end of the Pliocene and the beginning of the Pleistocene periods. The tectonic activity of this graben divides the catchment into a northern and a southern part, according to the theory of tectonic dipoles (Figure 2), with the latter being lifted and the former sinking [32,68]. The southern part displays strong relief, while the northern displays milder topography. The Eastern flat plain of Sperchios River is characterized by very gentle slopes and forms an exceedingly long system of meanders (Figure 1b, red ellipse).

The western and southwestern parts of the Sperchios catchment are dominated by impermeable rocks (Paleocene–Eocene flysch of the East Pindus and Parnassos geotectonic zones). At the southeast part permeable calcareous rocks are prevalent (Middle Triassic–Jurassic massive dolomites and limestones of the Pelagonian zone) with fewer appearances of the Upper Cretaceous flysch (Beotia zone) [42,69,70]. The north and north-eastern part are formed by an ophiolitic complex in a shale-chert formation, while the central and lower part of the catchment is occupied by Neogene and Quaternary unconsolidated deposits. In total, almost 74% of the catchment's southern part is dominated by impermeable rocks that form a dense drainage network with high values of surface runoff (Figure 3) [32,69,71].

The four most significant sub-catchments of the Sperchios river catchment's southern part, namely (from west to east) Roustianitis, Inachos, Gorgopotamos, and Assopos, were selected for the study purposes. The selection was based on their intriguing characteristics (e.g., size, steep slopes etc.) and particularly the vast water volume contribution to the main watercourse flow (the last three homonym tributaries are considered equivalent to rivers) (Figure 1b) [42,70]. Roustianitis, Gorgopotamos, and Assopos display a southwest to northeast orientation, while Inachos is divided into two parts, with the eastern part orienting from south to north and the western part from southwest to northeast [72,73]. Four additional major sub-catchments, located at the north part of the Sperchios river catchment—namely (from west to east) Vitoliotis, Archanorema, Drimarorema, and Xirias—were also analyzed for comparison (Figure 1b).

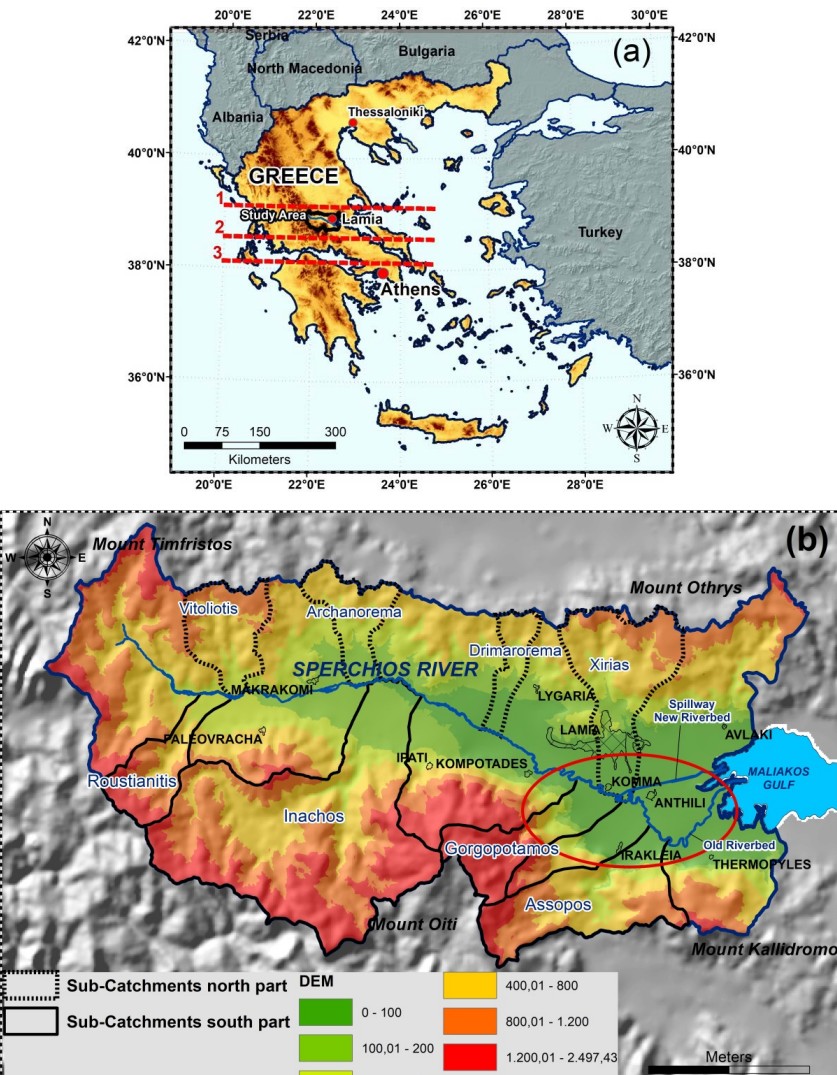

**Figure 1.** (**a**) The location of Sperchios river catchment in Central Greece (red lines and numbers 1, 2 and 3 showing the three areas of tectonic dipole's theory), (**b**) The morphology of the catchment and the distribution of the four main sub-catchments of its south part (Roustianitis, Inachos, Gorgopotamos and Assopos) and the corresponding sub-catchments of the north part (Vitoliotis, Archanorema, Drimarorema and Xirias). The red ellipse line shows the intense meandering part of River Sperchios.

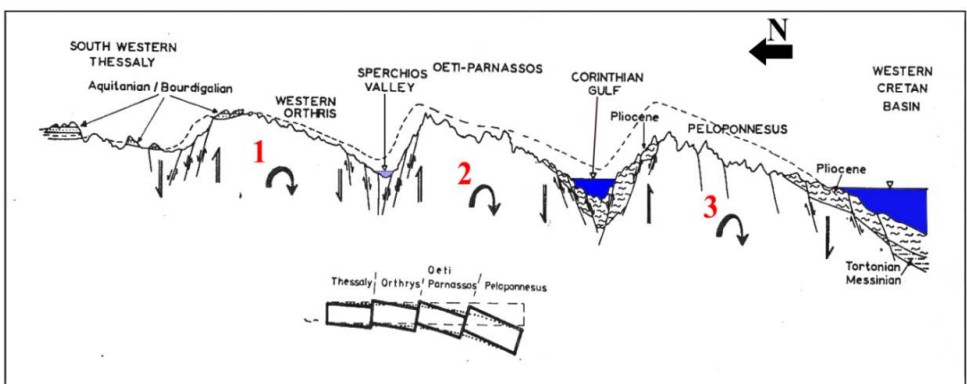

**Figure 2.** The tectonic activity of the area based on the tectonic dipole's theory (Mariolakos, 1976), red numbers 1, 2 and 3 showing the three parts of the tectonic dipole's theory (also appearing in Figures 1a and 3).

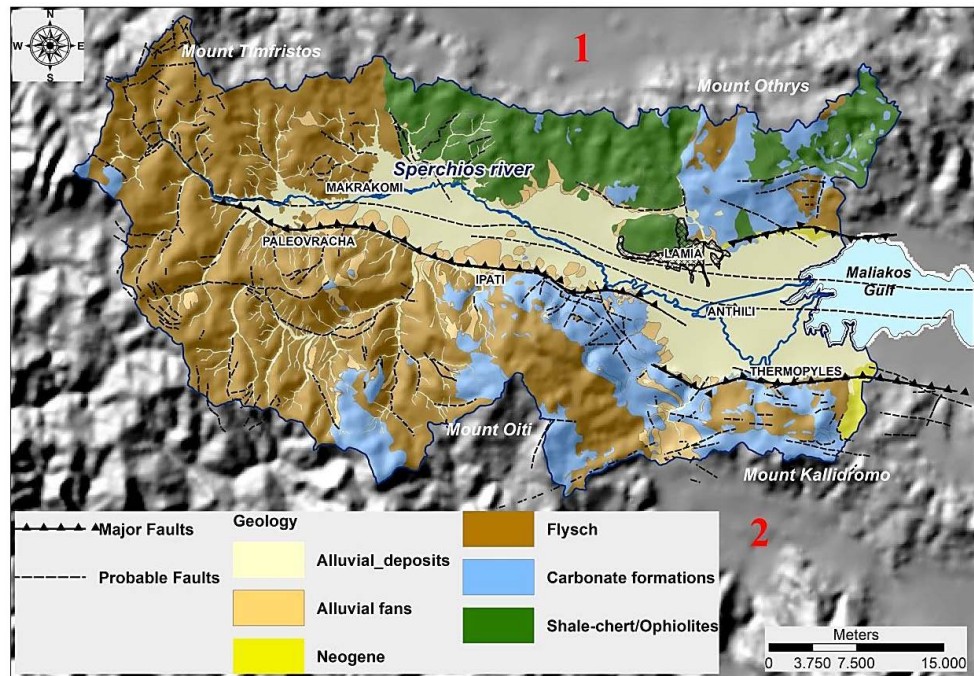

**Figure 3.** Simplified lithological map of the Sperchios river catchment, including faults and fracture systems derived from geological maps and satellite image interpretation (red numbers 1 and 2 showing the two areas of tectonic dipole's theory (also appearing in Figures 1a and 2)).

## 3. Data and Methodology

The methodology applied is schematically presented in Figure 4, and thoroughly analyzed in the following sections.

### 3.1. Geological and Lineament Mapping

The geological formations and lineaments digitization followed a manifold approach, comprising the interpretation of geological maps, field visits and a thorough literature review. Geological maps provided by the Institute of Geological and Mineral Exploration (1957–1991) of Greece (IGME, scale 1:50,000).

The lineament map developed was subsequently supplemented and finalized using high resolution (15 m) L7 images of different seasons (summer and winter), to decrease possible interpretation errors due to the adverse topography of the area. The comprehensive tectonic structure and its correlation with the morphometry and drainage network evolution, especially at the uplifting south part was examined, by performing statistical and rose diagrams analysis [74].

Two Landsat 7 ETM+ images were used to integrate and validate the lineaments of the area. The images were acquired free of charge (Path 184/Row 33, acquisition days: April 13th and August 19th, 2016) via the United States Geological Survey portal (https://earthexplorer.usgs.gov/). The GIS-based morphometric analysis and RS data manipulation were performed using the ArcGIS (Environmental Systems Research Institute-ESRI, Redlands, CA, USA) and ENVI (L3Harris Geospatial Solutions, Pearl East Circle, Boulder, CO, USA) software, respectively.

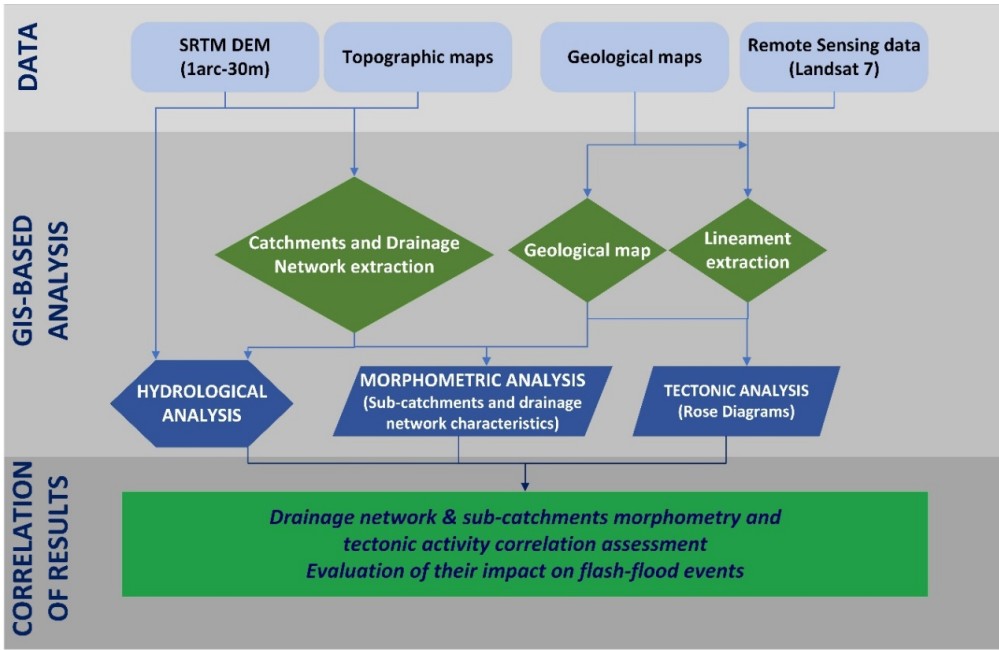

**Figure 4.** Methodology flowchart.

The methodology of the Landsat 7 ETM+ images processing for the extraction of linear features, in order to supplement and integrate to those derived from the geological maps, involved the pre-processing and processing phase. In the pre-processing stage, the images of L7 were radiometrically and atmospherically corrected and re-projected in the Greek Grid coordination system. In the processing stage, the fusion of multispectral bands (30 m) and the panchromatic band (15 m) was made using the Intensity Hue Saturation (IHS) technique (because it can preserve almost all of the spatial information that panchromatic has in the fused image) [75,76], in order to create a higher resolution (15 m) pan-sharpened image [77]. Subsequently, False Color Composites (FCCs) were created, and Principal Component Analysis (PCA) was implemented during the processing phase. The FCC image is an effective means for the visual interpretation of multispectral imagery, due to the sensitivity of the interpreter in color than greyscale brightness variations and the spectral characteristics of the utilized spectral bands. Numerous studies have used analogous FCC images for the extraction of lineament characteristics at other similar study areas in Greece [78–81]. The FCC created by using the Short Wave Infrared bands (7,5,3 as RGB) depicted more accurately the lineaments of the area [78,82,83].

The principal component (PC) transformation is a statistical technique of many variables that selects non-correlated linear compositions (eigenvectors) of a variable in such a way that each PC output has the minimum variance. In the multispectral images, this variable is related to the spectral response of various surface features. For the PCA, the six spectral bands (excluding the thermal band) of each image and the uncorrelated PCA bands were created by compressing and unmixing the spectral information of the original bands. Consequently, this transformation eliminates the redundancy of data, isolates noise, and then enhances the targeted information in the image. Several studies have been based on the PCA technique for the detection of lineaments, such as those of Walsh and Mynar [84], Paganelli et al. [85], and Adiri et al. [86] which showed that the PCA is very efficient in the identification of lineaments. The first component (PC1) corresponds to the brightness image while the rest contain spectral information related to other characteristics of the area. Additionally, in order to enhance the linear characteristics and the orientation contrast of the images (FCC and PC1), a high pass edge detection filter was applied (Figure 5a,b) [50,78,81]. PC1 reveals the information concerning the topography of the area, which, along with the edge enhancement, ideally highlights the linear elements of the image [87–90]. The final images of PC1 (along with the single panchromatic images of L7) and FCC were used for the photointerpretation process, and especially the extraction of the

small-scale linear features, which are not presented in the low-resolution geological maps. A detailed effort was made to exclude the linear features that do not correspond to geological structures, such as human-made linear features (roads, small runway, etc.) and land cover characteristics (crop-field boundaries, etc.). The final integrated map of the lineaments was accomplished by utilizing the geological maps of the area along with those extracted from satellite image interpretation.

### 3.2. Morphometric Analysis

Topographical delineation and morphometric analysis in the study area was based on eleven topographic maps, provided by the Hellenic Military Geographical Service (1971–1990) (HMGS—scale 1:50,000, contour interval 20 m), and a 30-m Digital Elevation Model (DEM), acquired from the Shuttle Radar Topography Mission (SRTM) dataset that was provided by the United States Geological Survey (USGS). The SRTM 1 Arc-Second Global elevation data were processed from raw C-band radar satellite data [91–94]. Specifically, the topographic maps were used for the delineation of the drainage network as they provide more detailed and accurate information about the actual shape and length of the streams (especially for the definition of the starting points of the first order streams, and the flowpaths at the meandering part in the flat plain area), while the SRTM DEM was used in all the other analysis. DEM pre-processing involved the correction of errors—i.e., filling of depressions. The robust handling of DEM depressions is essential for reliable hydrological analysis. Then, using the SRTM-DEM and GIS raster operations, the water divides, the estimation of the morphological features and the DEM-derived geospatial characteristics (e.g., slope angle, slope aspect, etc.) were extracted [42]. Additionally, the corrected SRTM-DEM was used for the computation of the Spatially Distributed Unit Hydrograph (SDUH).

For each sub-catchment, the drainage network characteristics, such as stream order, number and length, were identified, based on well-established theoretical principles of the Strahler classification system [10]. Drainage network ordering can convey information regarding its development and extent within a catchment. The morphometric parameters of each sub-catchment—i.e., (a) area ($A$), (b) perimeter length ($P$), (c) length ($Lb$) [95], (d) width ($Br$), (e) relative relief ($H$) (the difference between the maximum and minimum elevation), (f) relief ratio ($RH$) (the ratio of the catchment relief to the catchment length) were measured and analyzed (Table 1). The $e$ and $f$ factors reveal the influence of the catchment's relief on the drainage network formation [12]. In an attempt to assess the impact of tectonic processes on drainage network development, the (a) drainage density ($D$) (the ratio of the total stream length to the area of the watershed) and (b) drainage frequency ($F$) (the ratio of the total number of streams to the area of the catchment) indices were also calculated (Table 1).

Both coefficients provide information related to the surface runoff potential, slope steepness and rock permeability conditions. Moreover, in order to investigate the catchments' geometries, shape tilting and hydrological conditions, several other parameters were calculated as well: (a) the form factor ($Rf$) (the ratio of the watershed area to the square of the catchment length; indicates the flow intensity of the drainage network—catchments with high $Rf$ experience more significant peak flows in a shorter time), (b) the circularity ratio ($Rc$) (the ratio of the watershed area to the area of a circle with the same perimeter to the watershed; it is influenced by the frequency of the streams, the slopes and the geological structure), (c) the elongation ratio ($Re$) (the ratio of the watershed diameter, projected as a circle, to the watershed length; attributes the proportion of the catchment that has been elongated by tectonic activity, principally) and (d) the sinuosity index ($C$) (the ratio of the stream length to the catchment length; defines how straight/direct is a stream; completely straight/direct channels have a sinuosity index value close to 1.0, while low meandering streams have a sinuosity index value of 1.25–2.0 and high meandering streams >2.0 or more) [96] (Table 1).

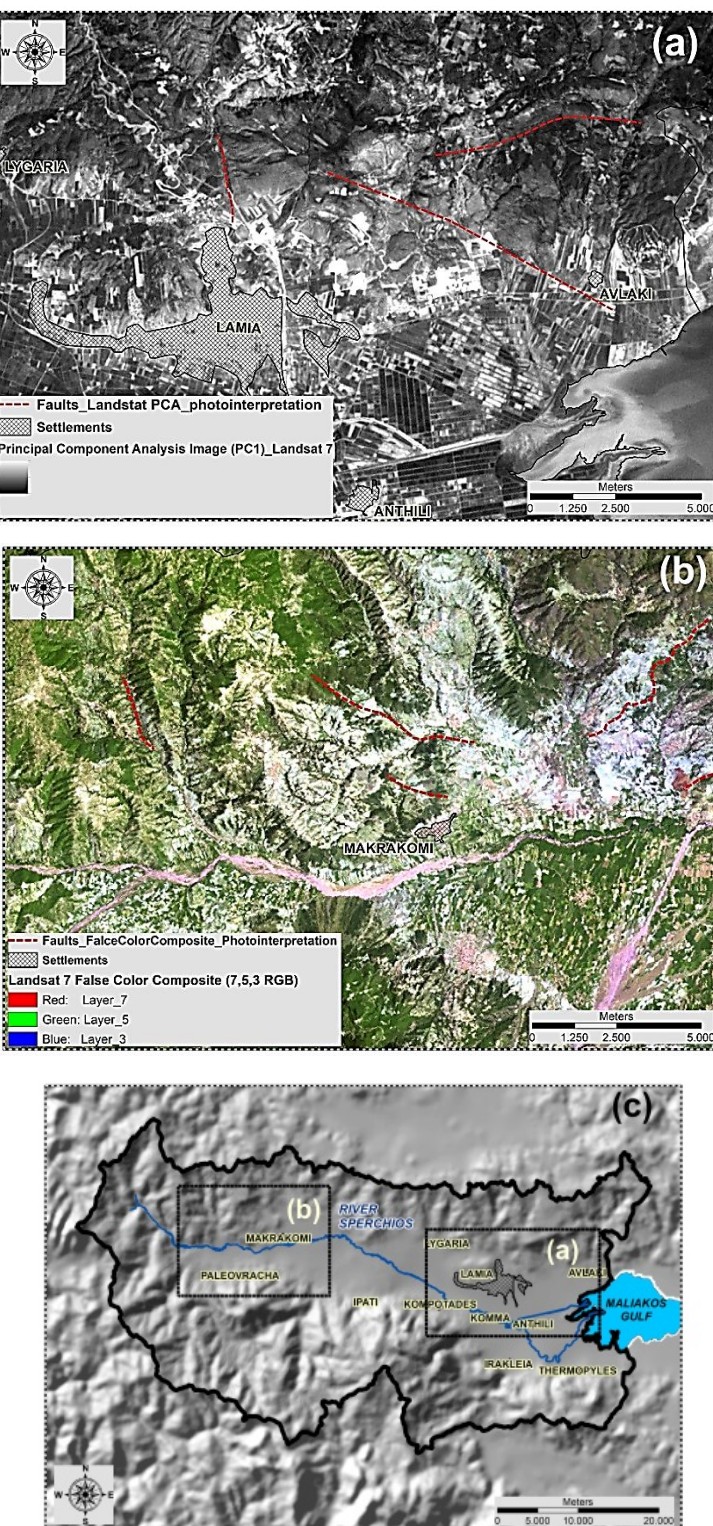

**Figure 5.** (**a**) Principal Component Analysis (Principal Component 1), (**b**) False Color Composite image (7,5,3 RGB), of Landsat 7 ETM+ for the validation and completion of the lineament map of the area (red lines presenting lineaments recognized and digitized through the visual interpretation of the images) and (**c**) an image showing the location of (**a**,**b**) zoomed areas.

**Table 1.** The morphometric characteristics and parameters of the selected sub-catchments

| Morphometric Parameters | Roustianitis | Inachos | Gorgopotamos | Assopos | Vitoliotis | Archanorema | Drimarorema | Xirias |
|---|---|---|---|---|---|---|---|---|
| | South Part | | | | North Part | | | |
| Drainage density (km/km$^2$) | 2.33 | 3.47 | 2.69 | 2.32 | 3.74 | 3.81 | 3.70 | 2.08 |
| Drainage frequency (n/km$^2$) | 3.86 | 8.45 | 5.87 | 4.64 | 9.74 | 10.21 | 8.75 | 4.12 |
| Area (km$^2$) (A) | 53.11 | 341.86 | 67.62 | 112.97 | 58.85 | 47.40 | 26.62 | 109.74 |
| Catchment length (km) (*Lb*) | 14.61 | 29.21 | 15.06 | 19.45 | 10.47 | 12.22 | 11.84 | 17.63 |
| Catchment width (km) | 3.64 | 11.70 | 4.50 | 5.81 | 5.62 | 3.88 | 2.25 | 6.22 |
| Perimeter (km) (P) | 39.67 | 104.58 | 52.90 | 58.95 | 36.28 | 38.54 | 30.84 | 57.63 |
| Maximum Elevation, H$_{max}$(m) | 1724.6 | 2293.6 | 2150.6 | 1804.5 | 1281.3 | 826.5 | 820.5 | 1085.51 |
| Minimum Elevation, H$_{min}$(m) | 240.5 | 90.0 | 14.0 | 8.0 | 215.0 | 100.5 | 30.8 | 12.2 |
| Total or Relative relief (m) | 1484.1 | 2203.6 | 2136.6 | 1796.3 | 1066.3 | 726.5 | 789.7 | 1073.5 |
| Median elevation, H$_{median}$(m) | 940.0 | 962.0 | 1320.0 | 680.5 | 670.2 | 420.0 | 285.5 | 593.1 |
| Mean Elevation, H$_{mean}$ (m) | 998.7 | 1155.4 | 1.089.7 | 946.94 | 747.4 | 452.6 | 456.1 | 540.0 |
| Mean slope (%) | 40 | 42 | 41 | 46 | 39 | 32 | 28 | 30 |
| Relief ratio (RH) (m km$^{-1}$) | 101.58 | 75.44 | 141.87 | 92.35 | 101.84 | 59.45 | 66.70 | 60.89 |
| Length-Width index (S) | 4.01 | 2.50 | 3.35 | 3.35 | 1.86 | 3.15 | 5.43 | 2.83 |
| Form factor | 0.25 | 0.40 | 0.30 | 0.30 | 0.54 | 0.32 | 0.19 | 0.35 |
| Circularity ratio | 0.42 | 0.39 | 0.30 | 0.41 | 0.56 | 0.4 | 0.35 | 0.42 |
| Elongation ratio | 0.21 | 0.72 | 0.62 | 0.62 | 0.54 | 0.64 | 0.49 | 0.67 |
| Drainage net. density index | 1.53 | 1.60 | 1.82 | 1.56 | 1.33 | 1.58 | 1.69 | 1.55 |
| Sinuosity index | 1.06 | 1.04 | 1.05 | 1.11 | 1.11 | 1.18 | 1.06 | 1.1 |

### 3.3. Drainage Network Analysis

To evaluate the drainage networks' and catchments' attributes, Horton's first (stream order) and second (stream length) laws were implemented [8,9], facilitating the estimation of various morphometric parameters [8,9].

Stream ordering is the first step taken in any drainage basin analysis and depends on basin shape, size, relief, geological and structural characteristics.

Horton's first law describes that the number of streams of successively lower orders forms a geometric development, where $\overline{Rb}$ = mean bifurcation ratio, K = maximum order of stream, $u$ = streams of a given order, Ni = ideal value for the number of streams of order u.

$$\mathrm{Ni} = \overline{\mathrm{R}}_{\mathrm{b}}^{\mathrm{K}-u} \tag{1}$$

The application of Horton's first law revealed useful information regarding the relationship between the number of the stream segments and the ordering of the drainage network. The law describes that, while stream order increases, the stream number decreases [26,97]. Deviations from their linear relationship imply that other parameters, such as tectonics and geological background, affect the drainage network development by influencing the stream order evolvement [12,27]. Bifurcation ratio (Rb) expresses the ratio of the number of streams of a given order (u) to those categorized in the next higher-order (u + 1) and indicates that, in the absence of strong geological formation and tectonic activity controls, the Rb shows a small variation in different regions. It is considered an important parameter, denoting the water carrying capacity and related flood potentiality of any catchment [26].

Horton's second law defines the mean stream length of each of the successive orders that tend to have a direct correlation to the stream length of the higher stream order u [9]. $\overline{Li}$ is the ideal value for the mean channel length of the order u and is calculated utilizing equation 2, while $\overline{L_1}$ is the mean channel length of the first order and $\overline{RL}$ is the mean length ratio (Table 2). Mean length was assessed using the regression of $\ln \overline{Lu} - \ln \overline{L_1}$.

$$\overline{\Sigma Li} = \overline{L_1}\,\overline{R_L}^{\,u-1} \tag{2}$$

**Table 2.** Horton's first and second law values for the four sub-catchments.

| RIVER | Stream Order (u) | Streams Number (Nu) | Bifurcation Ratio (Rb) | Mean Bifurcation Ratio ($\overline{Rb}$) | Ideal Streams Number (Ni) | Deviation | Streams Length (Lu) (km) | Mean Streams Length ($\overline{Lu}$) | Length Ratio (RL) | Mean Length Ratio ($\overline{RL}$) | Ideal Mean Streams Length (Li) | Deviation |
|---|---|---|---|---|---|---|---|---|---|---|---|---|
| Roustianitis | 1 | 156 | | | 194 | −38 | 70.2 | 0.45 | | | 0.45 | 0.0 |
| ΣNu = 205 | 2 | 36 | 4.33 | | 52 | −16 | 21.24 | 0.59 | 1.31 | | 1.07 | −0.48 |
| ΣLu = 123.87 (km) | 3 | 10 | 3.6 | 3.73 | 14 | −4 | 12.3 | 1.23 | 2.09 | 2.38 | 2.55 | −1.32 |
| | 4 | 2 | 5.0 | | 4 | −2 | 9.42 | 4.71 | 3.83 | | 6.07 | −1.36 |
| | 5 | 1 | 2.0 | | 1 | 0 | 10.71 | 10.71 | 2.27 | | 14.44 | −3.73 |
| | 1 | 2206 | | | 2365 | −159 | 661.8 | 0.3 | | | 0.3 | 0.0 |
| | 2 | 520 | 4.24 | | 648 | −128 | 254.8 | 0.49 | 1.63 | | 0.65 | −0.16 |
| Inachos | 3 | 118 | 4.4 | | 178 | −60 | 126.26 | 1.07 | 2.18 | | 1.41 | −0.34 |
| ΣNu = 2887 | 4 | 30 | 3.93 | 3.65 | 49 | −19 | 57.3 | 1.91 | 1.79 | 2.17 | 3.07 | −1.16 |
| ΣLu = 1186.02 (km) | 5 | 9 | 3.33 | | 13 | −4 | 27.0 | 3.0 | 1.57 | | 6.65 | −3.65 |
| | 6 | 3 | 3.0 | | 4 | −1 | 39.06 | 13.02 | 4.34 | | 14.44 | −1.42 |
| | 7 | 1 | 3.0 | | 1 | 0 | 19.8 | 19.8 | 1.52 | | 31.32 | −11.52 |
| Gorgopotamos | 1 | 304 | | | 345 | −41 | 100.32 | 0.33 | | | 0.33 | 0.0 |
| ΣNu = 397 | 2 | 71 | 4.28 | | 80 | −9 | 44.02 | 0.64 | 1.94 | | 0.92 | −0.28 |
| ΣLu = 123.58 (km) | 3 | 18 | 3.94 | 4.31 | 19 | −1 | 15.3 | 0.85 | 1.33 | 2.78 | 2.59 | −1.74 |
| | 4 | 3 | 6.0 | | 4 | −1 | 8.94 | 2.98 | 3.51 | | 7.24 | −4.26 |
| | 5 | 1 | 3.0 | | 1 | 0 | 13.0 | 13.0 | 4.36 | | 20.28 | −7.28 |
| Assopos | 1 | 405 | | | 425 | −20 | 141.75 | 0.35 | | | 0.35 | 0.0 |
| ΣNu = 524 | 2 | 91 | 4.45 | | 94 | −3 | 59.15 | 0.65 | 1.86 | | 0.94 | −0.29 |
| ΣLu = 261.95 (km) | 3 | 23 | 3.96 | 4.54 | 21 | +2 | 27.6 | 1.2 | 1.85 | 2.68 | 2.51 | −1.31 |
| | 4 | 4 | 5.75 | | 5 | −1 | 19.08 | 4.77 | 3.98 | | 6.74 | −1.97 |
| | 5 | 1 | 4.0 | | 1 | 0 | 14.37 | 14.37 | 3.01 | | 18.06 | −3.69 |
| | 1 | 437 | | | 580 | −143 | 120.5 | 0.28 | | | 0.28 | 0.0 |
| Vitoliotis | 2 | 104 | 4.2 | | 162 | −58 | 47.75 | 0.46 | 1.67 | | 0.49 | −0.03 |
| ΣNu = 573 | 3 | 24 | 4.33 | | 46 | −22 | 27.63 | 1.15 | 2.51 | | 0.85 | +0.30 |
| ΣLu = 219.8 km | 4 | 5 | 4.8 | 3.57 | 13 | −8 | 12.73 | 2.55 | 2.21 | 1.74 | 1.48 | +1.07 |
| | 5 | 2 | 2.5 | | 3 | −1 | 9.05 | 4.53 | 1.78 | | 2.57 | +1.96 |
| | 6 | 1 | 2.0 | | 1 | 0 | 2.32 | 2.3 | 0.51 | | 4.47 | −2.17 |
| | 1 | 361 | | | 541 | −180 | 98.65 | 0.27 | | | 0.27 | 0.0 |
| Archanorema | 2 | 94 | 3.84 | | 154 | −60 | 40.32 | 0.43 | 1.87 | | 0.51 | −0.00 |
| ΣNu = 484 | 3 | 22 | 4.27 | | 44 | −22 | 19.31 | 0.88 | 2.05 | | 0.98 | −0.10 |
| ΣLu = 180.8 km | 4 | 4 | 5.5 | 3.52 | 12 | −8 | 9.7 | 2.43 | 2.76 | 1.90 | 1.85 | −0.05 |
| | 5 | 2 | 2 | | 4 | −2 | 6.98 | 3.49 | 1.44 | | 3.52 | −0.03 |
| | 6 | 1 | 2 | | 1 | 0 | 5.84 | 5.84 | 1.67 | | 6.69 | −0.85 |
| Drimarorema | 1 | 179 | | | 222 | −43 | 55.7 | 0.31 | | | 0.31 | 0.0 |
| ΣNu = 233 | 2 | 42 | 4.26 | | 58 | −16 | 20.24 | 0.48 | 1.55 | | 0.97 | −0.49 |
| ΣLu = 98.4 km | 3 | 9 | 4.67 | 3.86 | 15 | −6 | 6.72 | 0.75 | 1.55 | 3.14 | 3.06 | −2.31 |
| | 4 | 2 | 4.5 | | 4 | −2 | 3.43 | 1.72 | 2.30 | | 9.60 | −7.88 |
| | 5 | 1 | 2.0 | | 1 | 0 | 12.31 | 12.31 | 7.18 | | 30.14 | −17.83 |
| Xirias | 1 | 346 | | | 352 | −6 | 116.07 | 0.34 | | | 0.34 | 0.0 |
| ΣNu = 452 | 2 | 81 | 4.27 | | 81 | 0 | 60.43 | 0.75 | 2.22 | | 0.88 | −0.13 |
| ΣLu = 228.5 km | 3 | 20 | 4.05 | 4.33 | 19 | +1 | 23.81 | 1.19 | 1.60 | 2.58 | 2.26 | −1.07 |
| | 4 | 4 | 5.0 | | 4 | 0 | 15.57 | 3.89 | 3.27 | | 5.84 | −1.95 |
| | 5 | 1 | 4.0 | | 1 | 0 | 12.6 | 12.60 | 3.24 | | 15.07 | −2.57 |

Stream length and mean stream length constitute important variables in a drainage basin analysis. Stream length indicates the successive stage of stream segment development [26,98]. Generally, the total length of streams acquires its maximum value if only the first order branches decrease as stream order increases. The deviations in this trend are indicative of discrepancies in geological characteristics (permeability of the rock formations) and tectonic activity. Mean stream length is related to the drainage network components of the catchment. Generally, higher stream orders indicate longer stream length and vice versa. The irregularities in mean length values over different orders suggest slope changes and specific geological setup or tectonic activity which by extension turn denotes abrupt changes in flow characteristics. Equation (3) (Horton 1932, 1945 [7,9]), Equaiton (4) (Horton 1932, 1945 [1,3]), Equaiton (5) (Apollov 1963 [95]), Equaiton (7) (Melton 1957 [99]), Equaiton (8) (Gregory and Walling 1973 [100]), Equaiton (9) (Schumm 1956 [14]), Equaiton (11) (Horton 1932 [7]), Equaiton (12) (Miller 1953 [101]), Equaiton (13) (Schumm 1956 [14]), Equaiton (14) (Luchisheva 1950 [102]), Equaiton (15) (Leopold et al. 1964 [25]), Equaiton (16) (Horton 1932 [7],

$$\text{Drainage density (km/km}^2) \rightarrow D = L_u/A \tag{3}$$

$$\text{Drainage frequency (n/km}^2) \rightarrow F = N/A \tag{4}$$

$$\text{Catchment length (km)} \rightarrow Lb \tag{5}$$

$$\text{Catchment width (km)} \rightarrow Br = A/Lb \tag{6}$$

$$\text{Total or Relative relief (m)} \rightarrow H = H_{max} - H_{min} \tag{7}$$

$$\text{Mean slope (\%)} \rightarrow BS = \Sigma Li\ d/A \tag{8}$$

$$\text{Relief ratio (RH) (m km}^{-1}) \rightarrow RH = H/Lb \tag{9}$$

$$\text{Length-Width index (S)} \rightarrow S = Lb/Br \tag{10}$$

$$\text{Form factor} \rightarrow Rf = A/Lb^2 \tag{11}$$

$$\text{Circularity ratio} \rightarrow Rc = 4\pi A/P^2 \tag{12}$$

$$\text{Elongation ratio} \rightarrow Re = 1.1w29A^{1/2}/Lb \tag{13}$$

$$\text{Drainage network density index} \rightarrow Co = 0.282P/A^{1/2} \tag{14}$$

$$\text{Sinuosity index} \rightarrow C = Lm/Lb \tag{15}$$

$$\text{Bifurcation ratio} \rightarrow Rb = Nu/Nu + 1 \tag{16}$$

where, $N_u$ = total number of streams, $L_u$ = total stream length (see Table 2), $Lm$ = main stream length, $Hmean - Hmin$ = the difference between the average catchment elevation and the elevation at the catchment outlet.

### 3.4. Tectonic and Drainage Network Correlation

With the goal of investigating the correlation of tectonic structure with the catchment's morphometric characteristics and their impact on the latter on its hydrological response, a morphometric, tectonic and hydrological analysis was made. More specifically, the four main sub-catchments of its southern part were chosen and, correspondingly, the four main sub-catchments of its northern part, served as benchmarks to compare the differences between the two sections of distinct morphometric and hydrological characteristics. Their selection was based on the inclusion of the main watercourse as the most essential within its perspective sub-catchment. Especially those of the south part with the more characteristic morphometry and their intense discharge rates have a significant impact on flooding occurrences [41,42,65,103].

The frequency and density rose diagrams of the drainage network branches and fault orientation were created to investigate their correlation with the tectonics of each sub-catchment. The initial consideration was to create rose diagrams of the faults that fall within each sub-catchment to check their relation with the order of the streams. Since in the Sperchios river case the sub-catchments are quite small and the tectonic activity in the area is a broader and more complicated phenomenon (most of the large faults extend through more than one sub-catchments), it was preferred the calculation of the rose diagrams of the whole basin would be better than limiting them to the sub-catchment level.

For every sub-catchment, the association between tectonics and sub-catchments' drainage network orientation and development process was examined by using linear element directions (lineaments and drainage network), which were estimated in the GIS environment. The lineaments were derived from the geological maps and the visual interpretation of the satellite images and were classified in two classes—the high-angle normal faults and the small-angle reverse faults (over-thrust, thrust)—using expert knowledge, extended fieldwork and literature review. Frequency (RF) and density (RD) statistical analyses were subsequently performed [104] utilizing the GEORIENT software to depict the relation between tectonic activity and drainage network orientation in the form of rose diagrams.

### 3.5. Analysis of Hydrological Response

Towards investigating the effect of the geomorphological characteristics on hydrological response, the runoff routing characteristics of all sub-catchments were estimated and compared. The meticulous examination of such characteristics led to the conclusion that a direct comparison could be made only in six of them, having comparable areas (pairwise, matching three sub-catchments from the southern part of Sperchios watershed to three sub-catchments from its northern part, as Roustianitis–Archanorema; Gorgopotamos–Drimarorema; Assopos–Xirias (Figure 1b)). A comprehensive investigation of the contribution of the geomorphological characteristics to the recurring flash-flood phenomena was attempted using the Spatially Distributed Unit Hydrograph (SDUH) method [105,106].

The SDUH is a unit hydrograph deriving from a spatially distributed unit excess rainfall, and it is compiled by analyzing the hydrologically corrected SRTM-DEM and the land cover of the catchment. It is computed from its characteristic time–area diagram, which is a graph of the cumulative catchment area whose time of travel is less or equal to a given value (isochronous method). The time–area diagram is calculated using a simple algorithm in a GIS that computes runoff travel times through the hillslopes (overland flow) and the channel network (channel flow) of the catchment, based on the flow velocity over each grid cell of the DTM [107]. Accordingly, the SDUH allows for the direct consideration of the geomorphologic characteristics of the catchment and, generally, of all spatially variable runoff routing parameters in the runoff routing process [107].

The methodology used in this study for the determination of the SDUH was implemented as follows: (i) The flow direction grids and the drainage networks for the studied catchments derived from the DTM analysis. (ii) The slope along the flow paths, and the drainage network grids, the flow velocity grids for the overland flow and the channel flow were generated based on the land cover. Flow velocity for overland flow ($V_o$) was computed as:

$$V_o = k \cdot J^{1/2} \tag{17}$$

Equation (16) derives from the Manning equation, assuming $k = R^{2/3}/n$. The $k$ coefficient is associated with land cover characteristics and the values corresponding to each land cover class were estimated according to McCuen [108]. The hydraulic gradient ($J$) was considered equal to the slope along the flow paths (mm$^{-1}$). The flow velocity for channel flow ($V_c$) was calculated using Manning's equation:

$$V_c = \frac{1}{n} \cdot R^{2/3} J^{1/2} \tag{18}$$

where the roughness coefficient (*n*) was set to an average value of 0.05 for all channels. The hydraulic radius (*R*) for each grid cell of the drainage network was determined by a power law function [109], which relates the hydraulic radius to the upstream area and provides a representation of the average behavior of the channel geometry

$$R = a \cdot A_d{}^b \tag{19}$$

where ($A_d$) is the cell's contributing area in square kilometers, (*a*) is a network constant and (*b*) a geometry scaling exponent. *Ad* is determined by the GIS flow accumulation routine, while *a* and *b* depend on the discharge frequency. In this application, the *a* and *b* parameters were set equal to 0.07 and 0.43, respectively, to correspond to normal floods [110]. The hydraulic gradient (*J*) was also considered equal to the slope along the flow paths (mm$^{-1}$). (iii) The two flow velocity grids (overland; channel flow) were overlaid and the final flow velocity grid for each catchment was calculated. The latter flow velocity grid is expressed in seconds per meter units—i.e., the time in seconds necessary for the water to cross a one-meter distance, facilitate the calculation of the travel time grids. (iv) The travel time grids were subsequently calculated, using the flow direction and the flow velocity grids as inputs in the GIS flow length routine. (v) Following the above steps, the time area diagrams for each catchment were subsequently derived by reclassifying the travel time grids to a five-minute time step (isochrone intervals equal to 5 min). Thereafter, the corresponding 3 h SDUHs were estimated based on the time area diagrams, assuming that 1 cm of excess rainfall was uniformly distributed over the watershed's surface.

Additionally, the water concentration times of the same sub-catchments were estimated using an empirical formula, categorized among the most suitable ones for European conditions [111,112]. This chosen method—e.g., in the lack of runoff gauges or at small catchments—is commonly used in Greece, where such cases are frequent, performing acceptably well (Equation (20) [113].

$$T_c = \frac{4A^{1/2} + 1.5L_b}{0.8(H_{mean} - H_{min})^{1/2}} \tag{20}$$

## 4. Results

### 4.1. Catchment Morphometric and Drainage Network Features Analysis

All sub-catchments have a dendritic type drainage network pattern, west–southwest to north–northeast orientation at the southern part, northwest to southeast and north–northeast to south–southwest orientation at the northern part. Inachos, Assopos (southern part) and Xirias (northern part) are the biggest catchments, covering areas of 341.86, 112.97, and 109.7 km$^2$, respectively. Gorgopotamos and Roustianitis (southern part), and Vitoliotis, Archanorema, and Drimarorema (northern part) are rather small catchments, covering areas of, 58.85, 67.62 and 53.11 km$^2$, respectively (Figure 6a–h, Table 1).

The upper regions of the Roustianitis, Inachos and Gorgopotamos catchments, as well as the Vitoliotis catchment (northern part), are dominated by the impermeable flysch formation, which influences the drainage network density.

Approximately 75% of the Roustianitis, Gorgopotamos, Assopos and Inachos catchments (southern part) occupy altitudes higher than 600–800 m, while 86.1%, 78.5%, 72.2% and 58.8% of Inachos, Roustianitis, Gorgopotamos and Assopos catchments, respectively, are characterized as mountainous with high total relief and very steep slopes [41]. On the contrary, at the northern part, apart from the Vitoliotis catchment, which is semi-mountainous with quite steep slopes, the other catchments display lower altitudes and gentle slopes.

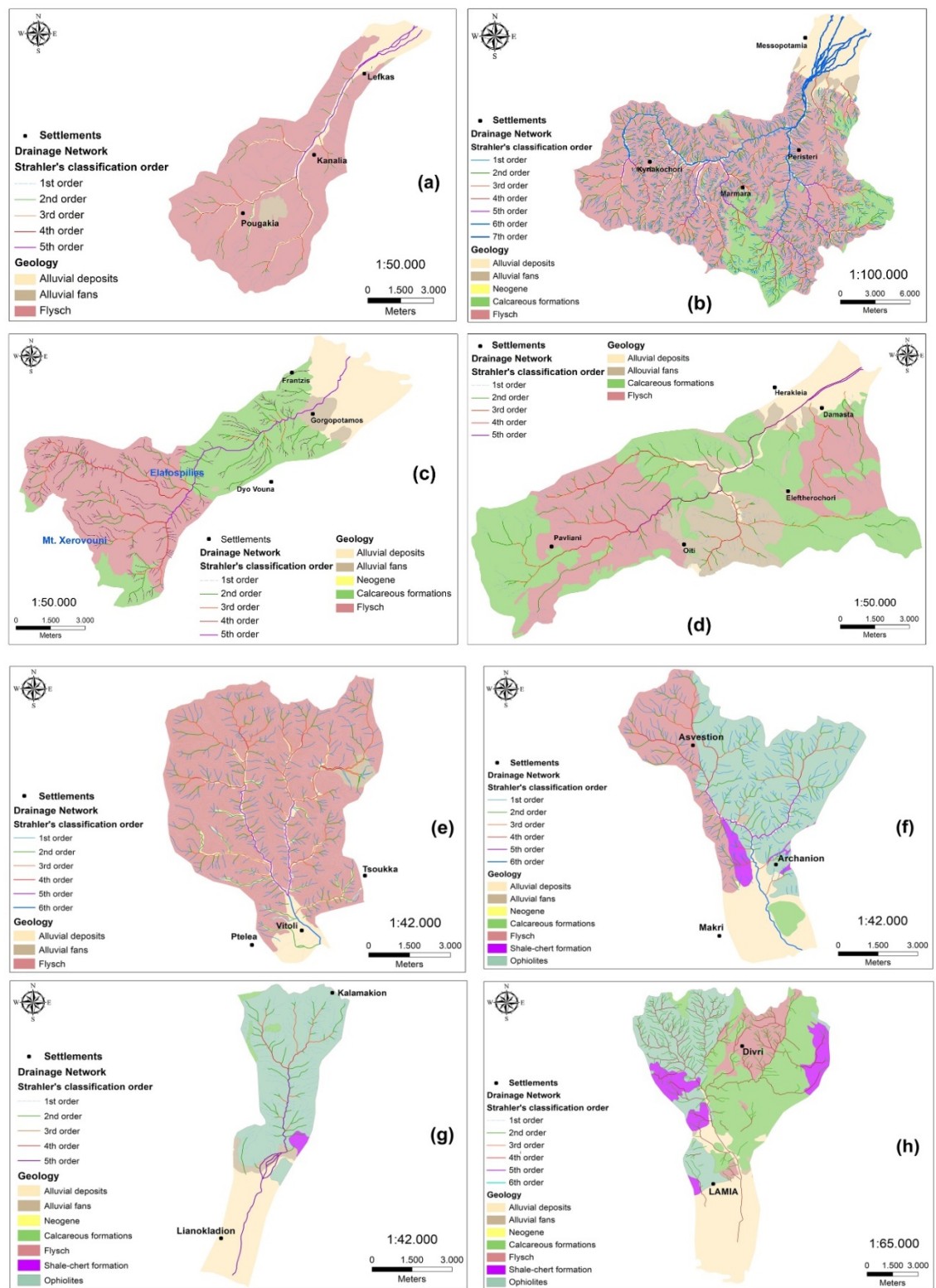

**Figure 6.** The geology and the classified drainage formation of (**a**) Roustianitis (**b**) Inachos, (**c**) Gorgopotamos, (**d**) Assopos, (**e**) Vitoliotis, (**f**) Archanorema, (**g**) Drimarorema and (**h**) Xirias catchments.

The implementation of Horton's laws shows that Roustiantitis, Gorgopotamos and Assopos, at the northern part, display negative deviation from the ideal values, for both stream number and length, indicating (especially the lengths) the existence of a drainage network which is in a young stage, affected mainly by the tectonic activity and considerably by the relief (steep slopes) and

geological formations (Figure 7a–h, Table 2). The north catchments, apart from Xirias, demonstrate the negative deviation of stream numbers from the ideal values. In the case of Vitoliotis, this is probably due to the geological background (flysch formation), while in Archanorema and Drimarorema it is related to geological and topographic characteristics. On the other hand, Xirias and Archanorema catchments show that the stream numbers and lengths, respectively, do not deviate from the ideal values, indicating a well-developed drainage network (intermediate stage), without any impact from the tectonic and geology. Contrary to the southern catchments, the northern catchment of Vitoliotis display a positive deviation of mean stream length values from the estimated ideal ones (Table 2), while Archanorema, Drimarorema and Xirias display negative values, a fact that reveals the impact of rock permeability (flysch at the northwestern part instead of carbonates and shale-chert formations at the northeastern part).

Furthermore, the bifurcation ratio index describes the degree of the structural complexity of a catchment and the influence of geological structure and tectonic activity on the drainage network development. When Rb ranges from 3 to 5, this indicates natural drainage system characteristics and a smaller influence of geological structures on the drainage networks. Within that range (3–5), lower values are considered to refer to less structural disturbances without drainage pattern distortion, while higher values can be indicative of high structural complexity and low permeability. Irregular values of Rb, either less than 3 or more than 5, are encountered in areas where geological control is dominant. The south catchments of Gorgopotamos and Assopos and the northern catchment of Xirias display the higher Rb values (>4) (Table 2), revealing the impact of tectonic activity at the southern and of the geological structure at the northern (northeastern) ones [12,114]. The same applies to the lower values (although >3.0) met on the remaining northern and southern catchments.

Drainage frequency primarily indicates the number of streams per unit area, which indirectly reveals the probable existence of conditions that favor runoff production (e.g., slope steepness, low rock permeability and increased precipitation depths). In general, the drainage frequency of Inachos and Gorgopotamos (south part) display greater than 5.0 values and, for Assopos, very close to 5.0, revealing the impact of impermeable geological formations (flysch), tectonic activity and high relief with steep slopes [41]. The same range of drainage frequency values (>5.0) appears in Vitoliotis, Archanorema and Drimarorema (north part), mainly due to the geological characteristics [115–117]. Contrariwise, Xirias and Roustianitis display lower values (<5.0) controlled by the presence of carbonate formations and the basin characteristics, respectively. Drainage density reveals information regarding surface runoff potential, ground surface steepness (including land cover), the degree of landscape formation (relief), rock permeability and susceptibility to erosion [12,92,117]. The drainage density (Dd) values of all sub-catchments are less than 5.0, a fact that is associated with coarse drainage network and geological characteristics [12]. The Dd of the north catchments, apart from Xirias, are higher than those of the south part, despite the extended appearance of flysch formations in the region (Figure 6a), which normally yield higher values of drainage density [42]. Hence, the low density is attributed to the slope gradient and the high relative relief, and the dense vegetation cover. The Inachos drainage density is calculated as 3.49, demonstrating the impact of the tectonic activity of the area (uplift of the Southern part of the Sperchios catchment), towards the development of few and short length streams (Figure 6b) [41,68]. Gorgopotamos and Assopos also demonstrate low density values (2.69 and 2.30), which are related to the presence of the permeable carbonate rocks, leading to the development of a sparse drainage network (Figure 5c,d) [54,71]. The overall conclusion is that the ongoing tectonic uplift of the southern part of the Sperchios catchment, along with the steep relief and the appearance of carbonate formation at the eastern part of the basin (cases of Xirias, Gorgopotamos and Assopos), contributes to the development of catchments with few streams and short lengths (Figure 6a–h, Table 2) [38,41].

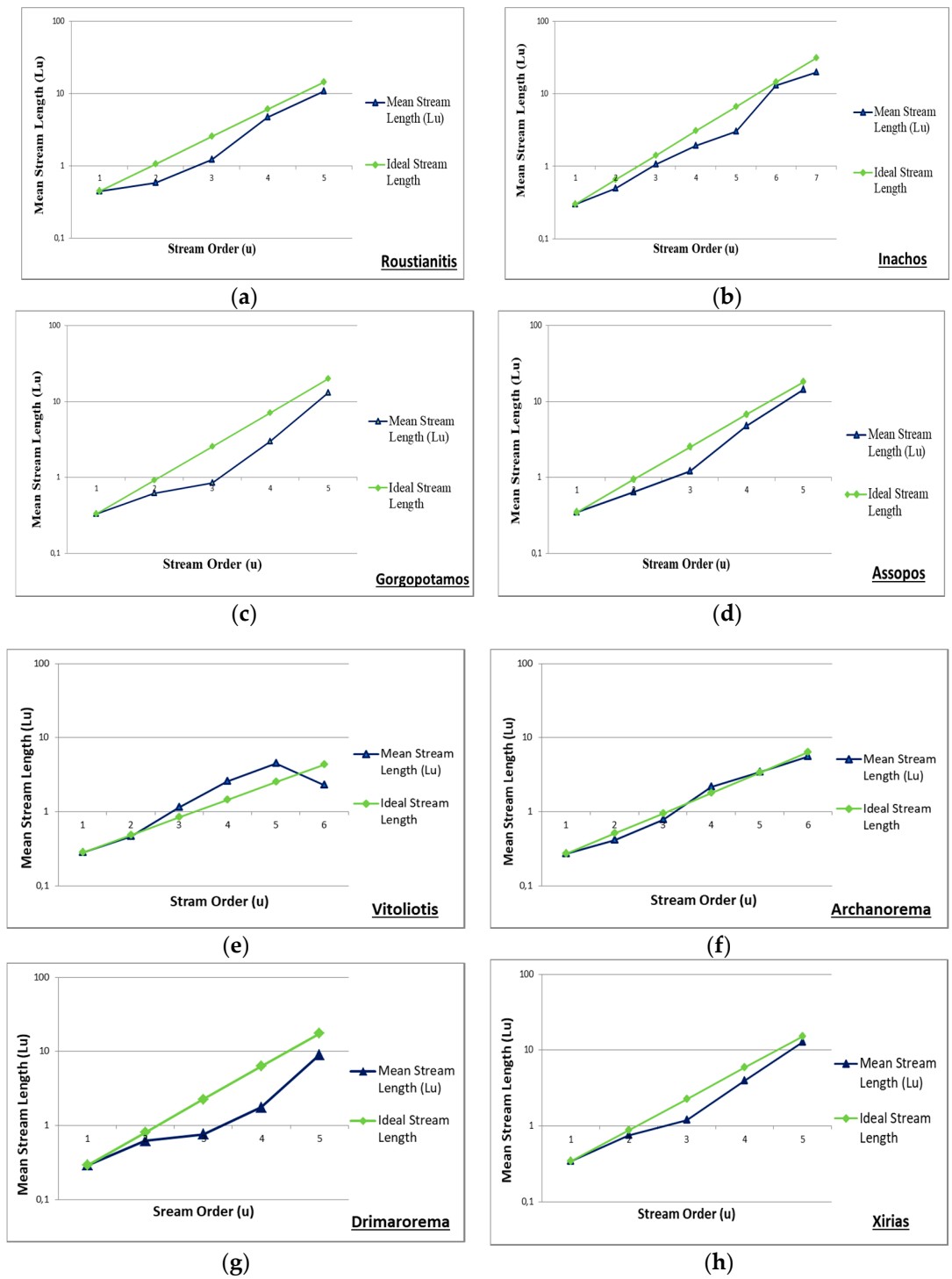

**Figure 7.** Diagrams showing the streams' mean length deviation from the ideal values of Horton's law for the four south sub-catchments, (**a**) Roustianitis, (**b**) Inachos, (**c**) Gorgopotamos, (**d**) Assopos and the four north sub-catchments (**e**) Vitoliotis, (**f**) Archanorema, (**g**) Drimarorema and (**h**) Xirias.

The Roustianitis catchment displays high mean elevation (998.7 m); total relief (1484.1 m); mean slope (40%); relief ratio (101.58 m km$^{-1}$) and slope index (95.75 m km$^{-1}$) values. The length–width ratio appears equally high, characterizing catchments with elongated shape, affected by the neo-tectonic activity of the area. Inachos shows the highest mean elevation (1155.4 m) and total relief (2203.6 m) values of all the study area sub-catchments. The mean slope (42%), relief ratio (75.44 m km$^{-1}$) and slope index (72.49 m km$^{-1}$) values are also high. The form factor, circularity ratio and length–width

index values are low (0.40, 0.39 and 2.50, respectively), contrary to the elongation ratio value, which is considered high (0.72). In general, low form factor values imply the presence of elongated basins with less side flow for a shorter duration and high main discharge [116,118,119]. The latter indicate significant elongation of the sub-catchment, which is affected by tectonic processes. Commonly, elongation ratio values close to or less than 0.5 indicate tectonically active regions [14,120]. Gorgopotamos morphology is characterized by a high maximum elevation value (2136.61 m) and steep slopes. The elongation ratio is rather high (0.62) but close to 0.5, indicating that the catchment has an elongated shape, affected by the tectonic activity (uplift) in the area. Finally, Assopos morphology is characterized by high maximum elevation (1796.25 m), mean elevation (946.94 m) and mean slope values (46%; especially at the southern and eastern parts of the catchment) values. The circularity and elongation ratio is estimated as 0.41 and 0.62 (close to 0.5), respectively, indicating that Assopos is an elongated catchment affected by the tectonic activity and geology of the area [42,70]. The highly elongated catchments have low circularity ratio values between 0.40 and 0.50 and they are less efficient in routing/draining discharge than the circular catchments [1], but the very steep slopes and the geological background of the south part exaggerate this point. In general, the elongation and circularity ratio of the north catchments is a little higher than the southern catchment due to the small size of the basins and the activity of local faults.

The moderate the high sinuosity index values of the southern catchments' drainage network density indexes (Roustianitis 1.53, Inachos 1.60, Gorgopotamos 1.82 and Assopos 1.56) along with their steep slopes, we depicted the rapid water flow and the short concentration time (*Tc*) to the outlet of the catchments.

### 4.2. Tectonic and Catchment's Morphometry-Drainage Network Correlation

The frequency and density analyses of the faults depict that the high angle normal faults have an east–west orientation, while the small-angle reverse faults display south–southeast to north–northwest orientation. The fault orientation rose diagrams of the drainage network indicate that, at Roustianitis, the first and second order streams display a southeast to northwest and east to west orientation, respectively, following the high angle normal faults orientation (Figure 8). Contrarily, the third, fourth, and fifth order branches display a northeast to southwest orientation, following the structure of low angle reverse faults (upthrusts, overthrusts). At Inachos the first order streams show east to west orientation, following the high angle normal faults one. The third and fourth order branches display south–southeast to north–northwest orientation following the structure of the low angle reverse faults. The second, third, sixth and seventh order branches display a northeast to southwest orientation, following the normal faults, but they are mainly affected by the existing geological structure.

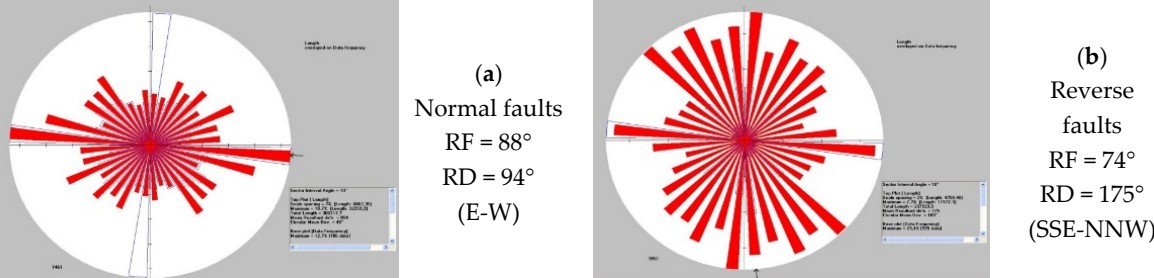

**Figure 8.** Rose diagrams of frequency (RF) and density (RD) for (**a**) high angle normal faults and (**b**) small-angle reverse faults (overthrust, thrust).

At Gorgopotamos the first to fourth order streams dominantly display an east to west orientation, following the high angle normal faults, while the main watercourse (fifth order) displays a northeast to southwest orientation and seems to be affected either by high angle normal faults or other older

faults of the area. Finally, at the Assopos sub-catchment the streams of first to fourth order appear to have an east to west and southeast to northwest orientation, following the high angle normal faults, as well as the main watercourse path (fifth order), displaying northeast to southwest orientation, affected either by high angle normal faults or other older faults of the area (Figure 9).

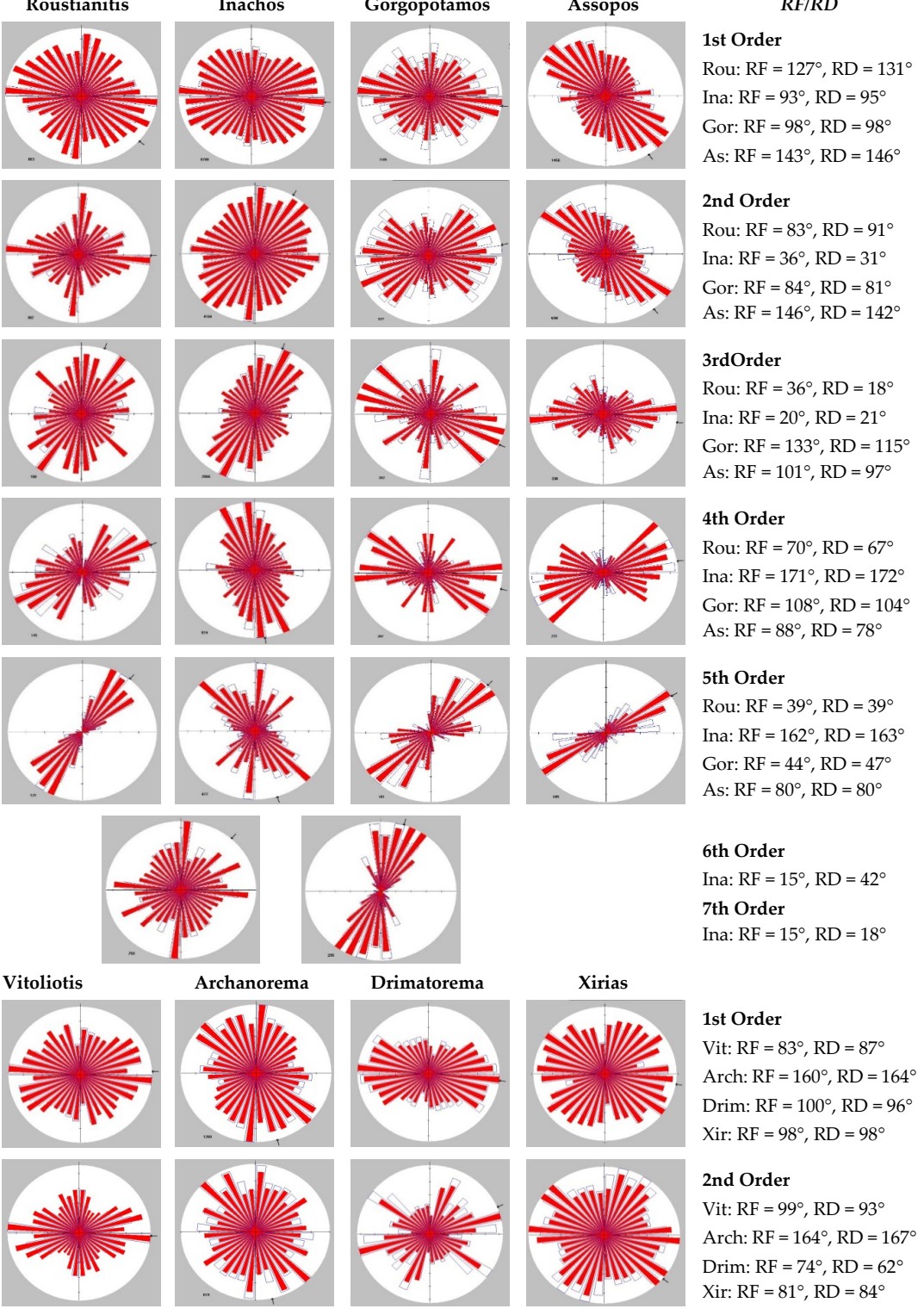

**Figure 9.** *Cont.*

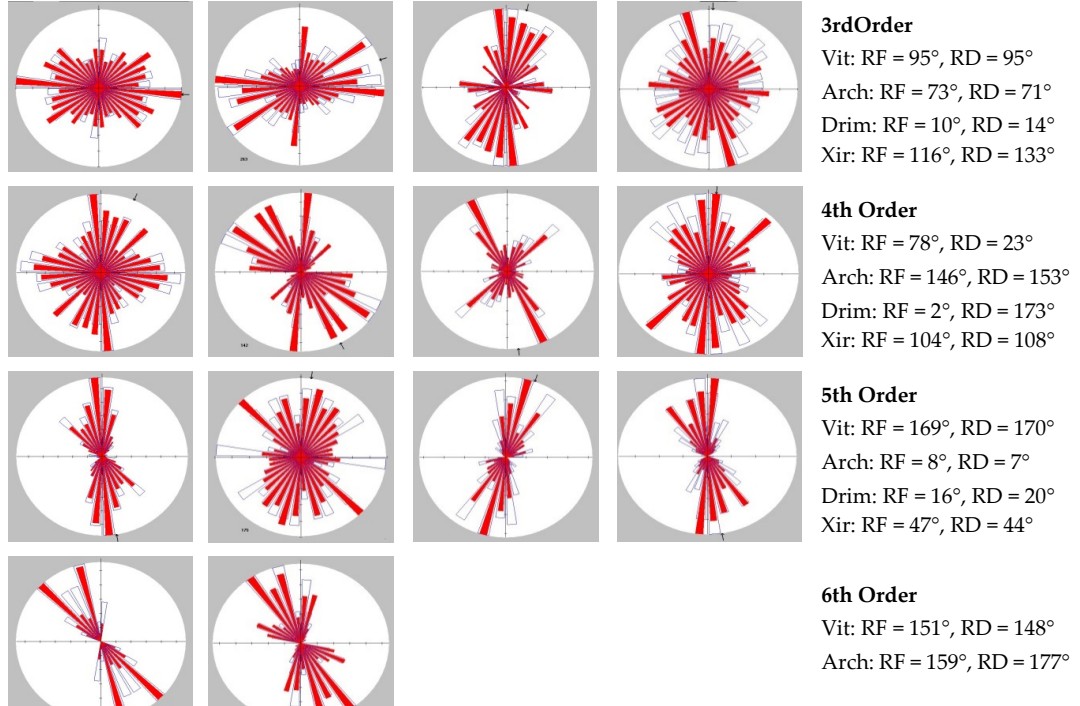

**Figure 9.** Rose diagrams of frequency (RF) and density (RD) of the eight sub-catchments' drainage network with their mean angle range.

Moreover, at Vitoliotis catchment, the streams of the fourth, fifth and sixth order are influenced mainly by low angle reverse faults or older faults of the area, such as the of first, second and fourth up to sixth order steams of Archanorema catchments, the streams of third, fourth and fifth order streams (main branch) of Drimarorema and the main branch of Xirias river. All other stream orders follow the high angle normal faults, having an east to west orientation.

### 4.3. Water Flow Velocity and Time Concentration Estimation

According to the SDUH method, constant values of flow velocities for both overland flow and channel flow were used, corresponding to normal floods (1 to 2 years return period). The velocities were based on the geomorphologic and land cover characteristics, providing a sound basis for the comparison between the northern and southern part of Sperchios catchment. The time area curves and the corresponding SDUH for the examined catchments are presented in Figure 10.

The water concentration time graphs corresponding to the southern part of the Sperchios catchment (Figure 10a) are sharper than those representing the corresponding catchments of the northern part, indicating a much faster hydrological response. Accordingly, the contributing area to the streamflow of the main watercourse at each time interval is obviously larger in the catchments of the southern part due to the high relief, steep slopes and tectonic activity. These observations highlight the effect of the distinctive geomorphological and tectonic features of the catchment's southern part on the recursive flash flood events.

These effects are even clearer in the corresponding 3-h SDUHs (Figure 10b). The sub-catchments of the southern part are characterized by much sharper rising limps and eventually higher peak flows. A more unbiased comparison can be made between catchments with similar areas—e.g., as in the case of Assopos and Xirias catchments or Gorgopotamos, Roustianitis and Archanorema catchments. The peak time is also notably smaller in the southern catchments.

Regarding the empirical Giandotti formula, Roustianitis, Gorgopotamos and Assopos (southern part) yielded lower values than Archanorema, Drimarorema and Xirias (north part; Table 3). These values reveal faster flow towards the outlet of the catchments and thus a shorter water concentration time ($Tc$).

Inachos was the only exception, with moderate values being "extracted" by using the Giandotti method, probably due to the particular characteristics of the catchment (tilting process). The overall results attest to the frequent outburst of flooding phenomena, owing to the short concentration time of flowing water at the junction of the sub-catchments' torrents with the Sperchios River main watercourse [42,111].

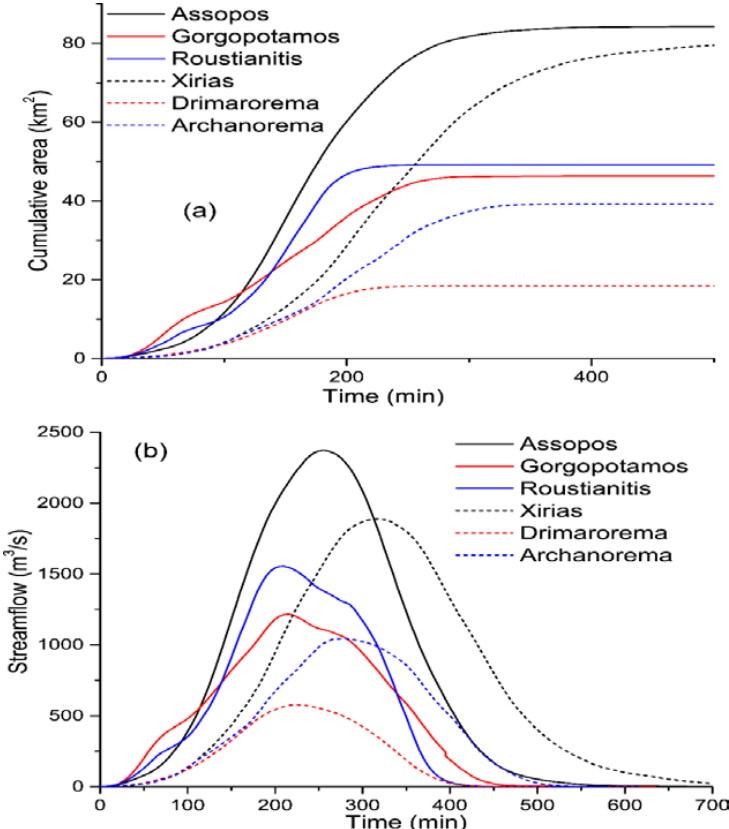

**Figure 10.** Graphs of (**a**) the time area curves of the investigated catchments and (**b**) the 3 h spatial distributed unit hydrographs, for the examined catchments. The first three catchments (solid lines) belong to the Southern part of Sperchios catchment and the other three (dashed lines) to the Northern part.

**Table 3.** The calculated values of Giandotti formula for the comparable pairs of the southern and northern sub-catchments.

| Basin Pairs (South/North) | Roustianitis/Archanorema | Gorgopotamos/Drimarorema | Assopos/Xirias |
|---|---|---|---|
| Giandotti empirical formula results (h) | 2.32/3.05 | 2.11/2.29 | 2.92/3.68 |

## 5. Discussion

The overall outcomes of the present study regard the usefulness of the geospatial technologies in morphometric, tectonic and hydrological analyses of a catchment. The detection of lineaments can be easily performed in a GIS environment, using remote sensing data of Landsat satellite system, based on distinguished surface characteristics that could manifest a fault zone (for example, topography and drainage). Moreover, the SRTM-DEM and its derived products have been ascertained as the most important source for identifying catchment's characteristics and morphometric parameters [18,81,121]. Additionally, it should be noted that the 30-m spatial resolution of SRTM-DEM is appropriate for the scale of this study, as it is related to a large catchment, covering an area of approximately 1830 km². In smaller catchments, a more detailed and higher resolution DEM should be used. Additionally,

a special consideration is required in plain areas and for the definition of the first order streams' starting points, since SRTM does not provide a reliable representation in these cases.

The frequency and density rose diagrams show that the main orientation of the drainage network in the sub-catchments of the southern part is mainly E–W and less E/SE–W/NW and E/NE–W/SW, following the high angle normal faults. These results are in agreement with many studies that have been done in the area and concern the tectonic activity and its effect on the morphology of the area, such as those of Eliet and Gawthorpe [33], Apostolopoulos [22], Pechlivanidou et al. [34], Zovoili et al. [122], and Tsodoulos et al. [123]. According to Paraschou and Vouvalidis [72], the Inachos catchment "tilts" to the North regarding its symmetry axis orientation (thus displaying an asymmetric development), which divides its upper part into two distinct branches with different directions. Thus, the sub-basin of Inachos river constitutes an exception, by being additionally affected by older local faults, showing an NE rotation around a NE–SW axis. The phenomenon influences drainage network development. Oppositely, at the northern part, the drainage network displays more S/SE–N/NW than NE–SW orientation, following the low angle reverse faults (upthrust, overthrusts). An exception occurs in the western part, which is mainly affected by the normal faults or from older local faults with an E–W orientation [41].

The whole area, as Psomiadis et al. [41], Eliet and Gawthorpe [33], and Pechlivanidou et al. [34] also mention in their research, is affected by the tectonic activity, differently impacting its southern and northern part, given the effects of EW faults (graben) and the tectonic dipole activity. This leads to the lifting process of the southern part and the sinking and widening process of the northern part. This activity creates two sections of distinct geomorphological and hydraulic behavior. The southern part displays strong relief and very steep slopes (at a relatively small distance, an altitude difference of 300 m can be noted), while the northern one displays milder topography with lower altitudes. The eastern flat plain of the Sperchios River is characterized by very gentle slopes and forms an exceedingly long system of meanders. Gorgopotamos and Assopos (southern part) are the only tributaries that contribute directly to the downstream meandering section of the Sperchios River (Figure 1b), since Xirias (northern part) outflows into the existing spillway (new riverbed) at the northern part of the Maliakos gulf (Figure 1b) [41,65].

Moreover, it is indisputable that the area's geological background constitutes a fundamental parameter for the evolution processes of the drainage network. This is also analyzed in many comprehensive studies, such as those of Eliet and Gawthorpe [33], Zamani and Maroukian [124], Maroukian and Lagios [32], and Psomiadis et al. [65]. The western and southwestern part of the Sperchios catchment is dominated by impermeable rocks (flysch), forming a dense drainage network. In contrast, the south–southeastern part of the basin is controlled by permeable calcareous rocks, which form a sparse network. The frequency and density of the drainage network acquire mostly moderate to high values, mainly due to geological bedrock and the recent ongoing neo-tectonic activity of the area. Likewise, the highest density values are met in the presence of impermeable and semi-permeable geological formations.

Based on a comprehensive geospatial analysis of the morphometric characteristics of the eight main sub-catchments of Sperchios catchment, much useful information concerning their morphology was extracted. In particular, the sub-catchments at the southern part are longer and have relatively smaller width in proportion to their size, while the basins at the northern part display higher circularity. More specifically, the stream length analysis using Horton's laws showed that (in general) stream numbers and lengths of the southern catchments have negative deviations from the ideal values. The largest deviations appear at the highest stream orders, indicating that these streams are short because they are still in a youthful phase of development, affected primarily by the tectonic activity [12,29]. The stream lengths of the northern catchments of Xirias and Drimarorema also show negative deviations from the ideal values (especially the highest stream orders), a fact that indicates the more extensive and well-developed drainage network and the impact of the geological formation (carbonates) at the northeastern part of the area. Consequently, it can be concluded that the drainage

network follows the tectonic movements, the geology, and morphometry, and consequently, the stream branches in the south are shorter and dense, while in the north they are longer, and sparse.

The Gorgopotamos catchment is well-known for its famous homonym canyon. Its creation is related to the transition of the geological formations, from flysch to carbonate rocks, having different erodibility characteristics. The impermeable flysch formations and the steep slopes at the upper part of the catchment lead to increased surface water velocity. As the water reaches the more erodible limestone formation (Elafospilies, Gorgopotamos catchment place name, Figure 6c) with extremely high (thus corrosive) speed, it forms a vast retrogressive erosion development, progressing towards the creation of the canyon [41].

The Spatially Distributed Unit Hydrograph (SDUH) methodology and the Giandotti empirical formula results reveal shorter water-concentration times (Tc) towards the outlet of the catchments concerning the southern part of Sperchios River catchment, in comparison to the northern part.

The morphometric characteristics of the southern part, the impermeable geological composition (flysch) and the steep slopes of the sub-catchments, result in torrential hydrological conditions (rapid flow and the high volume of water discharge over a short period) during extreme rainfall events. Additionally, the strong meandering formations (downstream Kompotades village) of the Eastern part of Sperchios riverbed (which abruptly receives these high amounts of water) significantly slow down water velocity. Hence, the inability of the river to drain these extreme water volumes rapidly towards its outlet at the Maliakos Gulf is caused, provoking an increase in water level and, ultimately, overflow, and, by extension, flash-flood events. The Sperchios catchment is an area extremely vulnerable to flood incidents, having suffered very intense and catastrophic flash-floods in the past, with the most notable ones being those of 1889, 1939, 1954, 1984, 1987, 1994, 1997, 2001, 2003, 2012 and 2015 (Figure 11). Many types of research, using conventional or innovative methods, have been conducted in the past and have highlighted the important issue of floods in the area, such as those of Psomiadis [103], Stathopoulos et al. [125], Paparrizos and Maris [126], and Bournas et al. [127]. Therefore, exploring flood preventive measures and improved intervention strategies should be identified as one of the primary goals for Sperchios river catchment management [128,129]. These measures should mainly be aimed at the adoption of nature-based solutions, especially at the meandering plain area, without affecting the natural landscape. The high risk is also related to the intense anthropogenic activity of the vulnerable eastern part, which gathers all the basic infrastructures of the area (National highway and railway), the city of Lamia, and a highly productive agricultural activity.

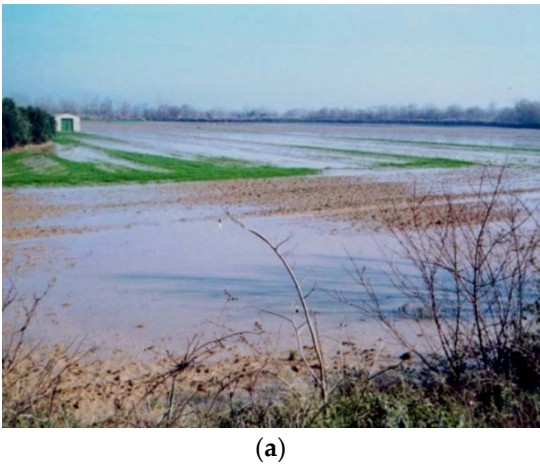 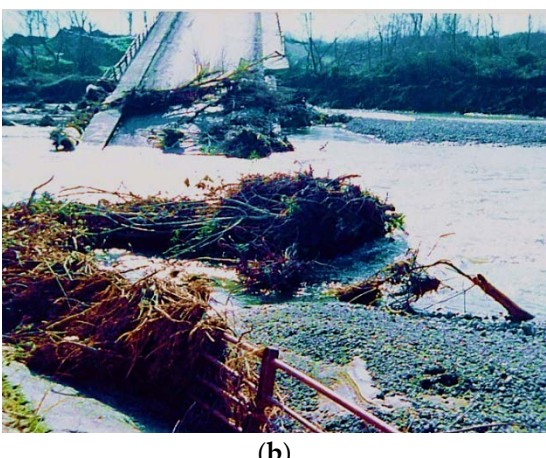

(**a**)                                        (**b**)

**Figure 11.** Photos demonstrating the severe flash-flood events that occurred (**a**) on February 1st 2015 and (**b**) on January 14th 1997. In the latter case, extended destruction occurred, and six bridges of the river collapsed.

The identical dynamic tectonic activity and intense geomorphological anaglyph of many watersheds of the Greek mainland depict the necessity of pursuing analogous thorough research efforts for the prevention of flash-flood phenomena. Numerous studies have been carried out in recent decades, trying to reveal and underline the necessity to investigate this dynamic state of different regions, such as those of Argyriou et al. [12], Charizopoulos et al. [29], and Ntokos [30].

Of course, similar analyses presuppose the existence of comprehensive and highly accurate geospatial data, the acquisition of which might be a potential constraint, although nowadays this problem tends to be surpassed by their increasing and affordable availability. Moreover, it must be noted that, in many cases, the tectonic activity can also affect infiltration and groundwater recharge, driving water to flow towards subsurface layers. In the present study, this perspective was disregarded (it constitutes a fruitful topic that can be analyzed in the near future), given the complexity of the underground system and the fact that it was considered a matter in contrast to the other parameters involved.

## 6. Conclusions

The overall examination and association between the sub-catchments' morphometry, drainage networks, and tectonics, can lead to useful conclusions, regarding their impact on the hydrological response of an area. In this study, the geospatial analysis of the tectonic activity and the quantitative description of Sperchios River catchment morphometric characteristics, along with their correlation to its hydrological response and the impact on flash-flood events were performed, using detailed RS data and GIS analysis, as well as field surveys.

The thorough morphometrical analysis was made using detailed topographical data and a medium- to high-resolution SRTM-DEM. The tectonic analysis utilized a combination of geological maps and supplementary data, such as properly processed L7 images and thorough literature analysis. The correlation of the tectonic activity with the morphometric evolution of the basin (especially of the southern part) was developed through the comparison of rose diagrams. Then, a hydrological analysis was performed utilizing a GIS-based SDUH and an empirical formula, in order to delineate the extremely high susceptibility of the area to flash flood events. Finally, the strong correlation of tectonics with morphometry and their direct impact on the frequent flash flood phenomena was depicted, revealing the necessity for upgraded management plans regarding flood mitigation and protection of Sperchios river catchment.

The similarity of Sperchios catchment tectonic activity and morphometric forms with several other regions of Greece, having the same distinguish features and geodynamic regime, reflects the importance of the current research and the essentiality to build on it, expanding in other comparably vulnerable areas.

**Author Contributions:** Conceptualization, E.P., K.X.S. and N.C.; methodology, E.P., K.X.S., N.C. and N.E.; software, E.P., K.X.S. and N.C.; data analysis, E.P., K.X.S., N.C. and N.E.; resources, E.P., K.X.S. and N.C.; writing—original draft preparation, E.P., K.X.S., N.C. and N.E.; supervision, E.P.; and fieldwork, E.P. and N.C. All authors have read and agreed to the published version of the manuscript.

**Funding:** This research received no external funding.

**Acknowledgments:** E.P. would like to thank the Greek State Scholarship Foundation for financially supporting the research activities conducted in this manuscript, as part of his PhD dissertation.

**Conflicts of Interest:** The authors declare no conflict of interest.

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
