# Peer review of "Investigating the Correlation of Tectonic and Morphometric Characteristics with the Hydrological Response in a Greek River Catchment Using Earth Observation and Geospatial Analysis Techniques"

_geosciences, doi:10.3390/geosciences10090377_

Round 1

Reviewer 1 Report

This paper exploits the correlation of tectonic and morphometric characteristics by using earth observation and geospatial analysis techniques.
The authors are using morphometric analysis to examine catchment dynamics and the hydrological and tectonic response to the development of the investigated catchments to support flood risk assessment.
Authors have provided a well written manuscript and would strongly encourage them to re-modify and organize it in a proper way to highlight the importance of their scientific interest.

Comments to be acknowledged:
• Line 30: Introduction lacks of detailed information regarding the background studies and the state of art of the use of morphometric indices in assessing tectonic activity and floods.
Something similar to what authors do for the lineament mapping via remote sensing in lines 67 to 82.
• Lines 83-103: This should be moved to the methodology, not in the introduction.
• Line 189: Correct to "Landsat 7 ETM+ "
• Lines 192-193: Any reason why this specific pan-sharpening technique was selected in relation to others? Use any supported bibliography if comparative analysis didn't take place.
• Line 199: Do you mean Figure 1b?
• Lines 198-199: Any references? How this FCC is proved to enhance lineaments?
• Lines 203-212: Why PCA was beneficial to use in relation to other approaches? Authors should also provide few more details in previous background studies of PCA applicability in lineament mapping.
• Line 263: The types of faults refers to the ones of geological maps or also the satellite derived ones? Where all those ones from visual interpretation of satellite images classified in situ? Please clarify with further details, else refer to linear elements derived by visual interpretation of satellite imagery and geological maps.
• Lines 283-286: Should be moved after line 276. Similarly for lines 289-323.
Separate appropriately and in a meaningful sequence the calculation of SDUH and the water concentration times.
• Line 334: The referred figures do not show the relation of formations vs drainage network density. Figure 7 shows simply the geological formations and Figure 8 the longitudinal profiles.
• Line 337: Which geomorphological features? This shows the geological formations using as basemap the Google Earth.
• Lines 344- 361: These lines seem to be interpretation from background studies. This is a results section. Those lines contain information that can be related with the outcomes of that study and need to be moved to discussion section to check the comparison of this study outcomes with previous studies.
• Lines 378-379: How is the presence of flysch influencing Vitoliotis catchment. Authors need to mention if they refer to the permeability characteristics of flysch formation. Provide further explanation.
• Lines 398-406: In Inachos basin the elongation ration is 0.72 and authors refer being affected by tectonic processes. In Gorgopotamos the same ration is 0.62 but authors refer to neotectonic activity.
• Line 405: What makes it neotectonic activity in opposed to the Inachos tectonic processes influence?
• Line 409: Similarly to comment at lines 398-406, Assopos has the same elongation value like Gorgopotamos, 0.62, but here authors refer to Assopos being affected by tectonic activity while Gorgopotamos by neotectonic activity. Explain/clarify the status and follow a consistent term.
• Lines 413-417: What about sinuosity index in relation to tectonic activity? Authors claim that low values highlight relatively straight/direct stream. Is that linearity reflecting the presence of tectonism directing them? Also, authors claim all sub-catchments to have absence of meanderisms but in lines 237-240 refer that values of that index over 1.5 imply to presence of meanderisms such as the ones mentioned in line 415. Please clarify.
• Lines 419-421: Are the orientations for the overall basin of Spercheios? Authors in the following lines compare them with the individual drainage basins. Why authors didn't create rose diagrams of the faults that fall within each basin to check their relation with the 1st, 2nd,3rd etc order drainage network?
• Line 422-423: Where is that shown? Figure 10 shows only normal and reverse faults rose diagrams? Where are the drainage network rose diagrams? Similarly for the following lines.
• Line 432: Which four catchments and why only four are acknowledged? What about the rest of the catchments? Also, rose diagrams of drainage network are missing.
• Line 480: Discussion section looks like a summary. Authors need to be critical about their outcomes and discuss thoroughly their interpretation and compare them to any existing previous studies.

Author Response

We have tried to address all your fruitful comments and moreover thorough editing in English was made.

More details you will find in the attached file.

Reviewer 2 Report

The manuscript by Psomiadis et al. titled ‘Investigating the correlation of tectonic and morphometric characteristics with the hydrological response in a Greek river catchment using earth observation and geospatial analysis techniques’ uses a GIS-based analysis to identify potential zones with high flood risk by correlating a watershed’s tectonic activity to its subbasins morphological parameters. The authors validate their hypothesis by quantifying the hydrologic response of different subbasins to conclude that the existing morphological and tectonic drivers significantly impact the response to hydrologic forcing. This is a novel concept and can certainly be useful to delineate potential factors that cause increased flood risk in tectonically active zones. I appreciate the authors effort to systematically analyze and quantify the drivers of flood risk. Please see below for minor suggestions to improve the manuscript further.

  1. What would be helpful is if the authors can list the exact steps followed at the end (preferably in the conclusions section) that can help in replicability of the analysis. Commenting on how this would applied across other landscapes and watersheds would increase the applicability of this manuscript.
  2. Please comment on the potential limitations of this approach in the absence of the data used in this analysis and discuss how would the approach change.
  3. Tectonic activity can also affect the infiltration and groundwater recharge by driving the flow through the subsurface. The hydrologic response quantified here is based on a relatively simplistic approach that lacks physicality. Please justify how the absence of subsurface processes in evaluating the hydrologic response does not influence the conclusions drawn from this analysis.
  4. The systems morphology impacts the hydrologic response, but extreme flood stressors in-turn change the morphology. Would the temporal scales of these changes be long enough to not impact the tectonic activity/morphometric properties?

Author Response

We have tried to address all your fruitful comments.

More details you will find in the attached file.

Reviewer 3 Report

Introduction
1) Lines 63-64: SRTM is a unique source of higher accuracy data, mainly supporting morphometric...

Higher accuracy in this sense is very relative. The resolution is not that high and some hydrological features are very much generalised at this resolution and data. But it is sufficient to be used for hydrological analysis at the catchment level knowing the limitations of the data.

Methods
2) You have mentioned under 3.1 how you used (overall) the DEM and GIS for deriving the morphological features. In 3.2.2, how did you exactly perform the GIS processing to produce these information? You only described here the parameters you derived but not HOW you derived them?

3) Every parameter and GIS processing you used for deriving e.g. the length and width of the streams, as well as the catchment areas, which you used for the computation of all morphometric parameters, are relative to the methods of how you processed the data. Did you automatically generate them using GIS and the DEM or you delineated the watersheds from the topographic map? Describe. If you have extracted them using hydrological analysis in GIS, then how exactly you performed this (particularly deriving streams and watersheds)?

4) Table 1
a) Correct drainage density for Inachos, Gorgopotamos, Assopos, Vitoliotis. For Gorgopotamos and Vitoliotis, make sure you use the correct Total Lu (see comment below).

b) Lb for Drimarorema should be 2.25. If changed, value for Length width index should also be adjusted.

c) Check for decimals in Total or relative reliefs for Aesops and Xirias. Decimals in the relief ratio values should also be adjusted.

5) Table 2.
a) You used the notation Nu in equation 14 to refer to the ideal value for the number of stream, while in Table 2 you use it to to refere to stream number. Then you have the column called Ideal stream number. Use different notations for the two as this can be misleading.

The same is true for the Mean Lu and the Ideal Length ratio you refer in lines 256 to 257.

b) Total Lu (Stream length) written for Gorgopotamos and Vitoliotis should be be 181.58 km and 219.98 km, respectively. Make sure that corresponding drainage density is adjusted based on correction. c) Check mean bifurcation ratio (Mean Rb) values. All the decimals are off except for Archanorema and Xirias.

d) Ideal stream numbers. Correct for the following: Roustianiatis (all orders); Inachos (all except #2); Gorgopotamos (all, except #1); Assopos; Archanore (all except #2); Drimarorema (all, except #1 and #2). Vitoliotis has incorrect values for all. These values and results will influence the deviation computed. See allso the comment on using percentage for deviation (#5h). The new results will change the statements you made in the results and discussion sections.

e) Stream length (Lu) values for Xirias are missing.

f) Mean Lu of Assopos = 0.64. Mean Lus for all stream order in Archanorema and Drimarorema are not correct

g) Mean length ratio (RL) and Ideal mean stream length. Correct the value for Roustiantis, Inachos, Gorgopotamo and Assopos. Make sure to adjust values corresponding to Ideal length and the deviation. Recheck values for Ideal length values for Vitoliotis, Drimarorema, Xirias. Adjust deviation.

h) Instead of using % as deviation (for both ideal number and lengths) use difference instead. The percentage values can be misleading especially if you only have a few n. For example 2 out of 4, which gives 50% deviation from the original value, which seems to be large when just considering the percentage. How the numbers deviate from each other, as well as the trend in deviation, will be easier to understand in this case when using the difference in numbers.

6) Line 241
a) In the Law of stream number equation, what does the ideal value for the number of streams indicate in terms of the characteristics of the basin? How does this value differ from the number of stream?

b) In computing this, you need the bifurcuation (Rb) value. What does bifurcation indicate and what does the specific bifurcation value mean?

7) Line 255
The same comment for the second law on stream length. What will these values indicate? Fix the equations so that they are more understandable.

RESULTS
8) Lines 362 to 368
a) If you correct the computed values for the Mean Rb and the ideal stream numbers, you will notice that the positive deviation especially for Vitoliotis will be negative. There will be some positive, values but the difference between the Nu and the ideal stream number will not be big as what the initial computations show.

b) After you make the corrections in table 2 and analysed the deviations, what could be the reason why the deviation is high in Inachos, Vitoliotis and Archonorema compared with the other locations? What could have contributed to these huge deviations of greater than 100 particularly to first order streams?

c) How about with Xirias, which has the lowest deviation in the stream segments? What could have affected this?

d) Noticeable that as the stream order becomes higher, the difference/deviation is minimised. Why?

9) Difference in ideal and mean lengths (make sure that you have recomputed the values before answering this)
a) What do a positive and negative value indicate?

b) Why did Assopos get the highest positive deviation/difference and what could have affected this).

10) Lines 369-371
a) "... the south catchments present high values of mean bifurcation ratio exceeding the theoretically expected value (Table 2)" - what is the theoretically expected value in Table 2? is it the bifurcation ratio? Make it clear.

b) What do the mean bifurcation and bifurcation ratio values indicate in terms of the possibility of flooding or in terms of the characteristics of the drainage in your study area? With the values you got, which ranges from 3 to 4.x, can this indicate huge differences in the characteristics of the drainage patterns in all the study area? how?

11)lines 377-378
"In general, the drainage frequency and density values of the south catchments are lower and higher, respectively, from those of the northern part, except of Vitoliotis catchment due to the presence of flysch."

Drainage frequency and density values in southern catchment (Except for Xirias) were higher than those of the northern catchments (see Table 1)

12) lines 380 to 382
"The drainage density and frequency of Roustianitis is 2.33 and 3.86, respectively. Despite the extended appearance of flysch formations in the region (Figure 5a), which normally yields higher values of drainage density [35], these are relatively low. Hence, the low density is attributed to the high relief and the presence of very steep slopes.

This relates to my earlier question of how you generated and computed for the morphometric parameters, particularly the stream (see #3). If you use the SRTM data and GIS to automatically generate the streams, the low density can be attributed to the method as well as the resolution of the DEM data you used. See also my comment under discussion.

13) What is the relationship between the streamflow (peakier appearance) and the terrain characteristics in the south and northern part? How can this have affected the flow characteristics in the area?

DISCUSSION
14) Related to my question earlier, regarding the generation of streams and watershed, if you have derived them by using GIS and hydrological analysis, then, the number of streams and even the watershed size are affected by the threshold used. If you based your computations on the results from this, how can these factors affect the reliability of your entire results and the computed values you produced in Tables 1 and 2, as well as when estimating and analysing the water flow velocity and concentration, and morphometric analysis.

15) Lines 510 to 514
The statement here needs to be updated based on new calculations derived. Discuss comprehensively. What do the differences in the number of streams and stream lengths indicate generally? How will you compare your results to earlier studies?

16) What are the limitations of the computations you used in this study in deriving the results for the analyses. Do they provide the correct answers and how reliable are your results in relation to the equations and methods you implemented? Explain

Author Response

(The authors gave the same response as above.)

Round 2

Reviewer 1 Report

See attached file.

Author Response

(The authors gave the same response as above.)

Reviewer 3 Report

General comment:

Most of my concerns in the previous versions (particularly the mthods section) have been addressed in the paper. However, some entries in the computations in table 1 and most especially table 2 are still wrong. I suggest to correct these and look at the computations for the entire table. despite that i pinpoint them in my comments, look at everything to ensure that they are all correct. And since some of your results and discussions are connected with these results, make sure that you discuss them consistently with the changes you make.

Detailed comments:

1)Table 1.
a) Adjust decimals for drainage densities (see previous comment)
b) Drimarorema - Catchment width = 2.25

2)Table 2. Make sure that your calculations are correct. Decimals are still off in a lot of places. Recalculate all values so that you know if they are correct. This can be done easily in Excel.
a) Vitoliotis stream 1 . deviation is incorrect.
b) Arachnorema Stream 1 - Ni is incorrect.
c) Inachos: Mean Rb = 3.65; Ideal stream numbers and the deviations should be adjusted. The deviation from the ideal should not be this big.
d) Drimarorema: Stream 2's Rb =4.26; Therefore, the Mean Rb should be = 3.86. Adjust stream numbers and deviations based on this. g
e) Roustiantis: Ideal stream lengths (Li) for streams 3 to 5. Adjust decimals correctly. Deviation should also be adjusted based on new calculations.
f) Inachos: Ideal stream lengths (Li) for streams 5 to 7. Adjust decimals correctly. Deviation should also be adjusted based on new calculations.
g)Gorgorpotamos Ideal stream lengths (Li) for streams 3 to 5. Adjust decimals correctly. Deviation should also be adjusted based on new calculations.
h) Assopos:
i) Length ratio (RL) for streams 1 to 2. Mean length ratio should be adjusted a bit.
j) Ideal stream lengths (Li) for streams 3 to 5. Adjust decimals correctly. Deviation should also be adjusted based on new calculations.

k) Vitoliotis: Ideal strem lengths (Li) for streams 3 to 6. Adjust decimals correctly. Deviation should also be adjusted based on new calculations.
l) Archanorema & Drimarorema: Mmean length (Lu) and length ratios (RL) for all streams are incorrect. The mean length ratio is therefore incorrect and so as the Li and Deviations.
m) Xirias: Ideal strem lengths (Li) for stream 5. Correct Li and deviations for streams 2 to 5.

3)Lines 285 to 288
"Horton's second law defines the mean stream length of each of the successive orders that tend to have a direct correlation to the stream length of the higher stream order u [9], where Li is the ideal value for the mean channel length of the order u, Li is the mean channel length of the first order and
Geosciences 2020, RL is the mean length ratio (equation 2, Table 2). Mean length was assessed by using the regression of lnLu-ln L1."

In the sentence, equation 2 should refer to Li not to RL.

4) Lines 576 to 578
"More specifically, the stream length analysis using Horton’s laws showed that (in general) stream numbers and lengths of the southern catchments have negative deviations from the ideal values."

All streams have negative deviations regardless of being south or north. There were only 2 instances of having positive values, and the differences are not too much (i.e. 2 and 1). So what exactly do your results imply? Is it expected to have negative deviations in such results? Why?

5) Lines 578 to 585
"This indicates that these streams are short because they are still in a youthful phase of development, affected primarily by the tectonic activity [12,27]. The stream lengths of the northern catchments of Xirias and Drimarorema show also negative deviations from the ideal values, a fact that indicates the more extensive and well-developed drainage network and the impact of the geological formation (carbonates) at the northeastern part of the area. Thus, it can be concluded that the drainage network follows the tectonic movements, the geology, and morphometry, and consequently, the stream branches in the south are shorter and dense, while in the north are longer, and sparse.

This should be related instead to the amount of deviation between the streams and the ideal values not just being negative or positive devation. The deviations are dominantly negative. Those that received positive deviations are streams at lower orders but streams at the highest orders are the ones with the largest negative deviations. So try to explain it in this context.

6) Lines 391 to 394
"The implementation of Horton's laws shows that for Roustiantitis, Gorgopotamos, Assopos and Drimarorema shows that all stream orders display negative deviation from the ideal values, for both stream number and length, indicating the existence of an incomplete hydrographic network, affected by the relief (steep slopes), tectonic activity and geological formations (Figure 9 a-h, Table 2)."
Again, the stream numbers, regardless of the catchment are all negative (see comment #). The stream length however, varies. Why? Are differences in the stream length significant enough to support your statement? Which can better support your statement: the stream length or the stream number or both. Explain clearly.

Author Response

(The authors gave the same response as above.)

Round 3

Reviewer 1 Report

See attachment.

Author Response

We have tried to address all your fruitful comments.

We have highlighted with track changes the modifications we made to the manuscript to address the comments

More details you will find in the attached file.

Reviewer 3 Report

Still have incosistencies with the calculation in table 2. You have changed some fo the numbers but did not change all calculations dependent on them. See details in comment# 4)

1) Lines 156 to 164
"...Specifically, the topographic maps were used for the delineation of the drainage network as they provide more detailed and accurate information about the actual shape and length of the streams (especially for first order streams and at the meandering part in the flat plains), while the SRTM DEM was used in all the other analysis. DEM pre-processing involved the correction of errors, i.e. filling of depressions. Robust handling of DEM depressions is essential for reliable hydrological analysis. Then, using the DEM and GIS raster operations, the catchments outlines, the drainage network, the estimation of the morphological features and the DEM-derived geospatial characteristics (e.g. slope angle, slope aspect, etc.) were extracted [42]."

If you have not used the DEM for hydrologic analysis and delineation of drainage network, then why do you have to fill depressions? Filling depression is used to ensure a continuous network of streams/rivers in the DEM, that is done in stream and watershed delineation. If you have only use the SRTM to extract slope, aspect, and other topographic features, then, you should not have done this step.

2) Lines 169 to 172
"..The catchments’ and drainage network parameters were extracted from the topographic maps (delineation of the drainage network) and the DEM using the automatic geospatial calculations routines of the Geographical Information System (slope, aspect, stream length, catchment area, width, etc.)."

3) Extracted, does it mean streams have been digitised from the topo maps and watershed have been delineated manually?

4)Table 2.
Values for stream length (lu, length ratio nad mean length in Archanorema and Drimarorema are still miscalculated. Correct the values again and make sure you also adjust all computations (li and deviation) that rely on these. If one is wrong, everything is wrong

5) lines 424 to 425
"Drainage frequency primarily indicates the degree of slope steepness, rock permeability and surface runoff."

Wording. Drainage density actually indicates the number of streams per unit area (Drainage density=N/A). It doesn't indicate the degree of slope steepness, rock permeability and surface runoff but rather is related to these.

DISCUSSION
6) How will you relate the analyses with the SRTM data, which you used to extract topographic features, that you used for the calculation of some morphometric charactersitics (height, relative relief, slopeto the results you have gotten? How can the DEM affect the generation of streams and the lengths particularly the given resolution and most especially the parameters you used in producing them.

7) Lines 540-546
"The overall outcomes of the present study regard the usefulness of the geospatial technologies in the morphometric, tectonic and hydrological analysis of a catchment. The detection of lineaments can be easily performed in a GIS environment, using remote sensing data of Landsat satellite system, based on distinguished surface characteristics that could manifest a fault zone (for example 543 topography and drainage). Moreover, the SRTM-DEM and its derived products have been ascertained as the most important source for identifying drainage network and catchment’s characteristics [18,92,121]."

Yes, the SRTM is important. But, referring to my previous question, what are the limitations of the remote sensing data and the SRTM in the entire process and analysis of your results? This is not just producing the results from these datasets, but also knowing how your results can be affected by them and the GIS processes you employed in generating those results, which your analyses were based on.

Author Response

(The authors gave the same response as above.)

Round 4

Reviewer 3 Report

The paper has been improved according to the previous comments.